# Gradient Methods Provably Converge to Non-Robust Networks

**Gal Vardi**[*][†]
TTI-Chicago and Hebrew University
galvardi@ttic.edu

**Gilad Yehudai**[*]
Weizmann Institute of Science
gilad.yehudai@weizmann.ac.il

**Ohad Shamir**
Weizmann Institute of Science
ohad.shamir@weizmann.ac.il

## Abstract

Despite a great deal of research, it is still unclear why neural networks are so susceptible to adversarial examples. In this work, we identify natural settings where depth-2 ReLU networks trained with gradient flow are provably non-robust (susceptible to small adversarial $\ell_2$-perturbations), even when robust networks that classify the training dataset correctly exist. Perhaps surprisingly, we show that the well-known implicit bias towards margin maximization induces bias towards non-robust networks, by proving that every network which satisfies the KKT conditions of the max-margin problem is non-robust.

## 1 Introduction

In a seminal paper, Szegedy et al. [2013] observed that deep networks are extremely vulnerable to *adversarial examples*. They demonstrated that in trained neural networks very small perturbations to the input can change the predictions. This phenomenon has attracted considerable interest, and various attacks (e.g., Carlini and Wagner [2017], Papernot et al. [2017], Athalye et al. [2018], Carlini and Wagner [2018], Wu et al. [2020]) and defenses (e.g., Papernot et al. [2016], Madry et al. [2017], Wong and Kolter [2018], Croce and Hein [2020]) were developed. Despite a great deal of research, it is still unclear why neural networks are so susceptible to adversarial examples [Goodfellow et al., 2014, Fawzi et al., 2018, Shafahi et al., 2018, Schmidt et al., 2018, Khoury and Hadfield-Menell, 2018, Bubeck et al., 2019, Allen-Zhu and Li, 2020, Wang et al., 2020, Shah et al., 2020, Shamir et al., 2021, Ge et al., 2021, Daniely and Shacham, 2020]. Specifically, it is not well-understood why gradient methods learn *non-robust networks*, namely, networks that are susceptible to adversarial examples, even in cases where robust classifiers exist.

In a recent string of works, it was shown that small adversarial perturbations can be found for any fixed input in certain ReLU networks with random weights (drawn from a Gaussian distribution). Building on Shamir et al. [2019], it was shown in Daniely and Shacham [2020] that small adversarial perturbations (measured in the Euclidean norm) can be found in random ReLU networks where each layer has vanishing width relative to the previous layer. Bubeck et al. [2021] extended this result to general two-layers ReLU networks, and Bartlett et al. [2021] extended it to a large family of ReLU networks of constant depth. These works aim to explain the abundance of adversarial examples in ReLU networks, since they imply that adversarial examples are common in random networks, and in particular in random initializations of gradient-based methods. However, trained networks are clearly

---

[*]Equal contribution
[†]Work done while the author was at the Weizmann Institute of Science

36th Conference on Neural Information Processing Systems (NeurIPS 2022).

not random, and properties that hold in random networks may not hold in trained networks. Hence, finding a theoretical explanation to the existence of adversarial examples in trained networks remains a major challenge.

In this work, we show that in depth-2 ReLU networks trained with the logistic loss or the exponential loss, gradient flow is biased towards non-robust networks, even when robust networks that classify the training dataset correctly exist. We focus on the setting where we train the network using a binary classification dataset $\{(\mathbf{x}_i, y_i)\}_{i=1}^m \subseteq (\sqrt{d} \cdot \mathbb{S}^{d-1}) \times \{-1, 1\}$, such that for each $i \neq j$ we have $|\langle \mathbf{x}_i, \mathbf{x}_j \rangle| = o(d)$. E.g., this assumption holds w.h.p. if the inputs are drawn i.i.d. from the uniform distribution on the sphere of radius $\sqrt{d}$. On the one hand, we prove that the training dataset can be correctly classified by a (sufficiently wide) depth-2 ReLU network, where for each example $\mathbf{x}_i$ in the dataset flipping the sign of the output requires a perturbation of size roughly $\sqrt{d}$ (measured in the Euclidean norm). On the other hand, we prove that for depth-2 ReLU networks of any width, gradient flow converges to networks, such that for every example $\mathbf{x}_i$ in the dataset flipping the sign of the output can be done with a perturbation of size much smaller than $\sqrt{d}$. Moreover, the *same* adversarial perturbation applies to all examples in the dataset.

For example, if we have $m = \Theta(d)$ examples and $\max_{i \neq j} |\langle \mathbf{x}_i, \mathbf{x}_j \rangle| = \mathcal{O}(1)$, namely, the examples are "almost orthogonal", then we show that in the trained network there are adversarial perturbations of size $\mathcal{O}(1)$ for each example in the dataset. Also, if we have $m = \tilde{\Theta}(\sqrt{d})$ examples that are drawn i.i.d. from the uniform distribution on the sphere of radius $\sqrt{d}$, then w.h.p. there are adversarial perturbations of size $\tilde{\mathcal{O}}(d^{1/4}) = o(\sqrt{d})$ for each example in the dataset. In both cases, the dataset can be correctly classified by a depth-2 ReLU network such that perturbations of size $o(\sqrt{d})$ cannot flip the sign for any example in the dataset.

A limitation of our negative result is that it assumes an upper bound of $\mathcal{O}\left(\frac{d}{\max_{i \neq j} |\langle \mathbf{x}_i, \mathbf{x}_j \rangle|}\right)$ on the size of the dataset. Hence, it does not apply directly to large datasets. Therefore, we extend our result to the case where the dataset might be arbitrarily large, but the size of the subset of examples that attain exactly the margin is bounded. Thus, instead of assuming an upper bound on the size of the training dataset, it suffices to assume an upper bound on the size of the subset of examples that attain the margin in the trained network.

The tendency of gradient flow to converge to non-robust networks even when robust networks exist can be seen as an implication of its *implicit bias*. While existing works mainly consider the implicit bias of neural networks in the context of *generalization* (see Vardi [2022] for a survey), we show that it is also a useful technical tool in the context of *robustness*. In order to prove our negative result, we utilize known properties of the implicit bias in depth-2 ReLU networks trained with the logistic or the exponential loss. By Lyu and Li [2019] and Ji and Telgarsky [2020], if gradient flow in homogeneous models (which include depth-2 ReLU networks) with such losses converges to zero loss, then it converges in direction to a KKT point of the max-margin problem in parameter space. In our proof we show that under our assumptions on the dataset every network that satisfies the KKT conditions of the max-margin problem is non-robust. This fact may seem surprising, since our geometric intuition on linear predictors suggests that maximizing the margin is equivalent to maximizing the robustness. However, once we consider more complex models, we show that robustness and margin maximization in parameter space are two properties that do not align, and can even contradict each other.

We complement our theoretical results with an empirical study. As we already mentioned, a limitation of our negative result is that it applies to the case where the size of the dataset is smaller than the input dimension. We show empirically that the *same* small perturbation from our negative result is also able to change the labels of almost all the examples in the dataset, even when it is much larger than the input dimension. In addition, our theoretical negative result holds regardless of the width of the network. We demonstrate it empirically, by showing that changing the width does not change the size of the perturbation that flips the labels of the examples in the dataset.

## 2 Preliminaries

**Notations.** We use bold-face letters to denote vectors, e.g., $\mathbf{x} = (x_1, \ldots, x_d)$. For $\mathbf{x} \in \mathbb{R}^d$ we denote by $\|\mathbf{x}\|$ the Euclidean norm. We denote by $\mathbb{1}(\cdot)$ the indicator function, for example $\mathbb{1}(t \geq 5)$ equals 1 if $t \geq 5$ and 0 otherwise. We denote $\text{sign}(z) = \mathbb{1}(z \geq 0)$. For an integer $d \geq 1$ we denote

$[d] = \{1, \ldots, d\}$. For a set $A$ we denote by $\mathcal{U}(A)$ the uniform distribution over $A$. We use standard asymptotic notation $\mathcal{O}(\cdot)$ to hide constant factors, and $\tilde{\mathcal{O}}(\cdot)$ to hide logarithmic factors.

**Neural networks.** The ReLU activation function is defined by $\sigma(z) = \max\{0, z\}$. In this work we consider depth-2 ReLU neural networks. Formally, a depth-2 network $\mathcal{N}_{\boldsymbol{\theta}}$ of width $k$ is parameterized by $\boldsymbol{\theta} = [\mathbf{w}_1, \ldots, \mathbf{w}_k, \mathbf{b}, \mathbf{v}]$ where $\mathbf{w}_i \in \mathbb{R}^d$ for all $i \in [k]$ and $\mathbf{b}, \mathbf{v} \in \mathbb{R}^k$, and for every input $\mathbf{x} \in \mathbb{R}^d$ we have $\mathcal{N}_{\boldsymbol{\theta}}(\mathbf{x}) = \sum_{j \in [k]} v_j \sigma(\mathbf{w}_j^\top \mathbf{x} + b_j)$. We sometimes view $\boldsymbol{\theta}$ as the vector obtained by concatenating the vectors $\mathbf{w}_1, \ldots, \mathbf{w}_k, \mathbf{b}, \mathbf{v}$. Thus, $\|\boldsymbol{\theta}\|$ denotes the $\ell_2$ norm of the vector $\boldsymbol{\theta}$.

We denote $\Phi(\boldsymbol{\theta}; \mathbf{x}) := \mathcal{N}_{\boldsymbol{\theta}}(\mathbf{x})$. We say that a network is *homogeneous* if there exists $L > 0$ such that for every $\alpha > 0$ and $\boldsymbol{\theta}, \mathbf{x}$ we have $\Phi(\alpha\boldsymbol{\theta}; \mathbf{x}) = \alpha^L \Phi(\boldsymbol{\theta}; \mathbf{x})$. Note that depth-2 ReLU networks as defined above are homogeneous (with $L = 2$).

**Robustness.** Given some function $R(\cdot)$, We say that a neural network $\mathcal{N}_{\boldsymbol{\theta}}$ is $R(d)$-*robust* w.r.t. inputs $\mathbf{x}_1, \ldots, \mathbf{x}_n \in \mathbb{R}^d$ if for every $i \in [n]$, $r = o(R(d))$, and $\mathbf{x}' \in \mathbb{R}^d$ with $\|\mathbf{x}_i - \mathbf{x}'\| \leq r$, we have $\text{sign}(\mathcal{N}_{\boldsymbol{\theta}}(\mathbf{x}')) = \text{sign}(\mathcal{N}_{\boldsymbol{\theta}}(\mathbf{x}_i))$. Thus, changing the labels of the examples cannot be done with perturbations of size $o(R(d))$. Note that we consider here $\ell_2$ perturbations.

In this work we focus on the case where the inputs $\mathbf{x}_1, \ldots, \mathbf{x}_n$ are on the sphere of radius $\sqrt{d}$, denoted by $\sqrt{d} \cdot \mathbb{S}^{d-1}$, which generally corresponds to components of size $O(1)$. Then, the distance between every two inputs is at most $O(\sqrt{d})$, and therefore a perturbation of size $O(\sqrt{d})$ clearly suffices for flipping the sign of the output (assuming that there is at least one input with $\mathcal{N}(\mathbf{x}_i) > 0$ and one input with $\mathcal{N}(\mathbf{x}_i) < 0$). Hence, the best we can hope for is $\sqrt{d}$-robustness. In our results we show a setting where a $\sqrt{d}$-robust network exists, but gradient flow converges to a network where we can flip the sign of the outputs with perturbations of size much smaller than $\sqrt{d}$ (and hence it is not $\sqrt{d}$-robust).

Note that in homogeneous neural networks, for every $\alpha > 0$ and every $\mathbf{x}$, we have $\text{sign}\left(\Phi(\alpha\boldsymbol{\theta}; \mathbf{x})\right) = \text{sign}\left(\alpha^L \Phi(\boldsymbol{\theta}; \mathbf{x})\right) = \text{sign}\left(\Phi(\boldsymbol{\theta}; \mathbf{x})\right)$. Thus, the robustness of the network depends only on the direction of $\frac{\boldsymbol{\theta}}{\|\boldsymbol{\theta}\|}$, and does not depend on the scaling of $\boldsymbol{\theta}$.

**Gradient flow and implicit bias.** Let $S = \{(\mathbf{x}_i, y_i)\}_{i=1}^n \subseteq \mathbb{R}^d \times \{-1, 1\}$ be a binary classification training dataset. Let $\Phi(\boldsymbol{\theta}; \cdot) : \mathbb{R}^d \to \mathbb{R}$ be a neural network parameterized by $\boldsymbol{\theta}$. For a loss function $\ell : \mathbb{R} \to \mathbb{R}$ the *empirical loss* of $\Phi(\boldsymbol{\theta}; \cdot)$ on the dataset $S$ is

$$\mathcal{L}(\boldsymbol{\theta}) := \sum_{i=1}^n \ell(y_i \Phi(\boldsymbol{\theta}; \mathbf{x}_i)) . \tag{1}$$

We focus on the exponential loss $\ell(q) = e^{-q}$ and the logistic loss $\ell(q) = \log(1 + e^{-q})$.

We consider gradient flow on the objective given in (1). This setting captures the behavior of gradient descent with an infinitesimally small step size. Let $\boldsymbol{\theta}(t)$ be the trajectory of gradient flow. Starting from an initial point $\boldsymbol{\theta}(0)$, the dynamics of $\boldsymbol{\theta}(t)$ is given by the differential equation $\frac{d\boldsymbol{\theta}(t)}{dt} \in -\partial^\circ \mathcal{L}(\boldsymbol{\theta}(t))$. Here, $\partial^\circ$ denotes the *Clarke subdifferential*, which is a generalization of the derivative for non-differentiable functions (see Appendix A for a formal definition).

We say that a trajectory $\boldsymbol{\theta}(t)$ *converges in direction* to $\tilde{\boldsymbol{\theta}}$ if $\lim_{t \to \infty} \frac{\boldsymbol{\theta}(t)}{\|\boldsymbol{\theta}(t)\|} = \frac{\tilde{\boldsymbol{\theta}}}{\|\tilde{\boldsymbol{\theta}}\|}$. Throughout this work we use the following theorem:

**Theorem 2.1** (Paraphrased from Lyu and Li [2019], Ji and Telgarsky [2020])**.** *Let $\Phi(\boldsymbol{\theta}; \cdot)$ be a homogeneous ReLU neural network parameterized by $\boldsymbol{\theta}$. Consider minimizing either the exponential or the logistic loss over a binary classification dataset $\{(\mathbf{x}_i, y_i)\}_{i=1}^n$ using gradient flow. Assume that there exists time $t_0$ such that $\mathcal{L}(\boldsymbol{\theta}(t_0)) < 1$, namely, $y_i \Phi(\boldsymbol{\theta}(t_0); \mathbf{x}_i) > 0$ for every $\mathbf{x}_i$. Then, gradient flow converges in direction to a first order stationary point (KKT point) of the following maximum margin problem in parameter space:*

$$\min_{\boldsymbol{\theta}} \frac{1}{2} \|\boldsymbol{\theta}\|^2 \quad s.t. \quad \forall i \in [n] \ \ y_i \Phi(\boldsymbol{\theta}; \mathbf{x}_i) \geq 1 . \tag{2}$$

*Moreover, $\mathcal{L}(\boldsymbol{\theta}(t)) \to 0$ and $\|\boldsymbol{\theta}(t)\| \to \infty$ as $t \to \infty$.*

Note that in ReLU networks Problem (2) is non-smooth. Hence, the KKT conditions are defined using the Clarke subdifferential. See Appendix A for more details of the KKT conditions. Theorem 2.1 characterized the *implicit bias* of gradient flow with the exponential and the logistic loss for homogeneous ReLU networks. Namely, even though there are many possible directions $\frac{\theta}{\|\theta\|}$ that classify the dataset correctly, gradient flow converges only to directions that are KKT points of Problem (2). We note that such KKT point is not necessarily a global/local optimum (cf. Vardi et al. [2021]). Thus, under the theorem's assumptions, gradient flow *may not* converge to an optimum of Problem (2), but it is guaranteed to converge to a KKT point.

## 3   Robust networks exist

We first show that for datasets where the correlation between every pair of examples is not too large, there exists a robust depth-2 ReLU network that labels the examples correctly. Intuitively, such a network exists since we can choose the weight vectors and biases such that each neuron is active for exactly one example $\mathbf{x}_i$ in the dataset, and hence only one neuron contributes to the gradient at $\mathbf{x}_i$. Also, the weight vectors are not too large and therefore the gradient at $\mathbf{x}_i$ is sufficiently small. Formally, we have the following:

**Theorem 3.1.** *Let $\{(\mathbf{x}_i, y_i)\}_{i=1}^m \subseteq (\sqrt{d} \cdot \mathbb{S}^{d-1}) \times \{-1, 1\}$ be a dataset. Let $0 < c < 1$ be a constant independent of $d$ and suppose that $|\langle \mathbf{x}_i, \mathbf{x}_j \rangle| \leq c \cdot d$ for every $i \neq j$. Let $k \geq m$. Then, there exists a depth-2 ReLU network $\mathcal{N}$ of width $k$ such that $y_i \mathcal{N}(\mathbf{x}_i) \geq 1$ for every $i \in [m]$, and for every $\mathbf{x}_i$ flipping the sign of the output requires a perturbation of size larger than $\frac{1-c}{4} \cdot \sqrt{d}$. Thus, $\mathcal{N}$ is $\sqrt{d}$-robust w.r.t. $\mathbf{x}_1, \ldots, \mathbf{x}_m$.*

*Proof.* We prove the claim for $k = m$. The proof for $k > m$ follows immediately by adding zero-weight neurons. Consider the network $\mathcal{N}(\mathbf{x}) = \sum_{j=1}^m v_j \sigma(\mathbf{w}_j^\top \mathbf{x} + b_j)$ such that for every $j \in [m]$ we have $v_j = y_j$, $\mathbf{w}_j = \frac{2\mathbf{x}_j}{d(1-c)}$ and $b_j = -\frac{1+c}{1-c}$. For every $i \in [m]$ we have

$$\mathbf{w}_i^\top \mathbf{x}_i + b_i = \frac{2\|\mathbf{x}_i\|^2}{d(1-c)} - \frac{1+c}{1-c} = 1 \,,$$

and for every $i \neq j$ we have

$$\mathbf{w}_j^\top \mathbf{x}_i + b_j = \frac{2\mathbf{x}_j^\top \mathbf{x}_i}{d(1-c)} - \frac{1+c}{1-c} \leq \frac{2c}{1-c} - \frac{1+c}{1-c} = -1 \,.$$

Hence, $\mathcal{N}(\mathbf{x}_i) = v_i = y_i$ for every $i \in [m]$.

We now prove that $\mathcal{N}$ is $\sqrt{d}$-robust. Let $i \in [m]$ and let $\mathbf{x}_i' \in \mathbb{R}^d$ such that $\|\mathbf{x}_i - \mathbf{x}_i'\| \leq \frac{1-c}{4} \cdot \sqrt{d}$. We show that $\text{sign}(\mathcal{N}(\mathbf{x}_i')) = \text{sign}(\mathcal{N}(\mathbf{x}_i))$. We have

$$\mathbf{w}_i^\top \mathbf{x}_i' + b_i = \mathbf{w}_i^\top (\mathbf{x}_i' - \mathbf{x}_i) + \mathbf{w}_i^\top \mathbf{x}_i + b_i \geq -\|\mathbf{w}_i\| \cdot \|\mathbf{x}_i' - \mathbf{x}_i\| + 1 \geq \frac{-2}{\sqrt{d}(1-c)} \cdot \frac{\sqrt{d}(1-c)}{4} + 1 = \frac{1}{2} \,.$$

Also, for every $i \neq j$ we have

$$\mathbf{w}_j^\top \mathbf{x}_i' + b_j = \mathbf{w}_j^\top (\mathbf{x}_i' - \mathbf{x}_i) + \mathbf{w}_j^\top \mathbf{x}_i + b_j \leq \|\mathbf{w}_j\| \cdot \|\mathbf{x}_i' - \mathbf{x}_i\| - 1 \leq \frac{2}{\sqrt{d}(1-c)} \cdot \frac{\sqrt{d}(1-c)}{4} - 1 = -\frac{1}{2} \,.$$

Therefore, $\text{sign}(\mathcal{N}(\mathbf{x}_i')) = \text{sign}(v_i) = \text{sign}(y_i) = \text{sign}(\mathcal{N}(\mathbf{x}_i))$. $\qquad\square$

It is not hard to show that the condition on the inner products in the above theorem holds w.h.p. when $m = \text{poly}(d)$ and $\mathbf{x}_1, \ldots, \mathbf{x}_m$ are drawn from the uniform distribution on the sphere of radius $\sqrt{d}$. Indeed, the following lemma implies that in this case the inner products can be bounded by $\frac{d}{2}$, and can even be bounded by $\sqrt{d} \log(d)$ (See Appendix B for the proof).

**Lemma 3.1.** *Let $\mathbf{x}_1, \ldots, \mathbf{x}_m$ be i.i.d. such that $\mathbf{x}_i \sim \mathcal{U}(\mathbb{S}^{d-1})$ for all $i \in [m]$, where $m \leq d^k$ for some constant $k$. Then, with probability at least $1 - d^{2k+1} \left(\frac{3}{4}\right)^{(d-3)/2} = 1 - o_d(1)$ we have $|\langle \mathbf{x}_i, \mathbf{x}_j \rangle| \leq \frac{1}{2}$ for all $i \neq j$. Moreover, with probability at least $1 - d^{2k+1-\ln(d)/4} = 1 - o_d(1)$ we have $|\langle \mathbf{x}_i, \mathbf{x}_j \rangle| \leq \frac{\log(d)}{\sqrt{d}}$ for all $i \neq j$.*

## 4 Gradient flow converges to non-robust networks

We now show that even though robust networks exist, gradient flow is biased towards non-robust networks. For homogeneous networks, Theorem 2.1 implies that gradient flow generally converges in direction to a KKT point of Problem (2). Moreover, as discussed previously, the robustness of the network depends only on the direction of the parameters vector. Thus, it suffices to show that every network that satisfies the KKT conditions of Problem (2) is non-robust. Formally, we have:

**Theorem 4.1.** *Let* $\{(\mathbf{x}_i, y_i)\}_{i=1}^m \subseteq (\sqrt{d} \cdot \mathbb{S}^{d-1}) \times \{-1, 1\}$ *be a training dataset. We denote* $I := [m]$, $I^+ := \{i \in I : y_i = 1\}$ *and* $I^- := \{i \in I : y_i = -1\}$, *and assume that* $\min\left\{\frac{|I^+|}{m}, \frac{|I^-|}{m}\right\} \geq c$ *for some* $c > 0$. *Furthermore, we assume that* $m \leq \frac{d+1}{3(\max_{i \neq j} |\langle \mathbf{x}_i, \mathbf{x}_j \rangle|+1)}$. *Let* $\mathcal{N}_\theta$ *be a depth-2 ReLU network such that* $\theta$ *is a KKT point of Problem (2). Then, there is a vector* $\mathbf{z} = \eta \cdot \sum_{i \in I} y_i \mathbf{x}_i$ *with* $\eta > 0$ *and* $\|\mathbf{z}\| = \mathcal{O}\left(\sqrt{\frac{d}{c^2 m}}\right)$, *such that for every* $i \in I^+$ *we have* $\mathcal{N}_\theta(\mathbf{x}_i - \mathbf{z}) \leq -1$, *and for every* $i \in I^-$ *we have* $\mathcal{N}_\theta(\mathbf{x}_i + \mathbf{z}) \geq 1$.

**Example 1.** *Assume that* $c$ *(from the above theorem) is a constant independent of* $d, m$. *Consider the following cases:*

- *If* $\max_{i \neq j} |\langle \mathbf{x}_i, \mathbf{x}_j \rangle| = \mathcal{O}(1)$ *and* $m = \Theta(d)$ *then the adversarial perturbation* $\mathbf{z}$ *satisfies* $\|\mathbf{z}\| = \mathcal{O}(1)$. *Thus, in this case the data points are "almost orthogonal", and gradient flow converges to highly non-robust solutions, since even very small perturbations can flip the signs of the outputs for all examples in the dataset.*

- *If the inputs are drawn i.i.d. from* $\mathcal{U}(\sqrt{d} \cdot \mathbb{S}^{d-1})$ *then by Lemma 3.1 we have w.h.p. that* $\max_{i \neq j} |\langle \mathbf{x}_i, \mathbf{x}_j \rangle| = \mathcal{O}\left(\sqrt{d} \log(d)\right)$, *and hence for* $m = \Theta\left(\frac{\sqrt{d}}{\log(d)}\right)$ *the adversarial perturbation* $\mathbf{z}$ *satisfies* $\|\mathbf{z}\| = \mathcal{O}\left(\sqrt{\sqrt{d} \log(d)}\right) = \tilde{\mathcal{O}}\left(d^{1/4}\right) = o(\sqrt{d})$.

*Note that in the above cases the size of the adversarial perturbation is much smaller than* $\sqrt{d}$. *Also, by Theorem 3.1 and Lemma 3.1, there exist* $\sqrt{d}$-*robust networks that classify the dataset correctly.*

Thus, under the assumptions of Theorem 4.1, gradient flow converges to non-robust networks, even when robust networks exist by Theorem 3.1. We discuss the proof ideas in Section 5. We note that Theorem 4.1 assumes that the dataset can be correctly classified by a network $\mathcal{N}_\theta$, which is indeed true (in fact, even by a width-2 network, since by assumption we have $m \leq d$). Moreover, we note that the assumption of the inputs coming from $\sqrt{d} \cdot \mathbb{S}^{d-1}$ is mostly for technical convenience, and we believe that it can be relaxed to have all points approximately of the same norm (which would happen, e.g., if the inputs are sampled from a standard Gaussian distribution).

The result in Theorem 4.1 has several interesting properties:

- It does not require any assumptions on the width of the neural network.

- It does not depend on the initialization, and holds whenever gradient flow converges to zero loss. Note that if gradient flow converges to zero loss then by Theorem 2.1 it converges in direction to a KKT point of Problem (2) (regardless of the initialization of gradient flow) and hence the result holds.

- It proves the existence of adversarial perturbations for *every* example in the dataset.

- The *same* vector $\mathbf{z}$ is used as an adversarial perturbation (up to sign) for all examples. It corresponds to the well-known empirical phenomenon of *universal adversarial perturbations*, where one can find a single perturbation that simultaneously flips the label of many inputs (cf. Moosavi-Dezfooli et al. [2017], Zhang et al. [2021]).

- The perturbation $\mathbf{z}$ depends only on the training dataset. Thus, for a given dataset, the same perturbation applies to all depth-2 networks which gradient flow might converge to. It corresponds to the well-known empirical phenomenon of *transferability* in adversarial examples, where one can find perturbations that simultaneously flip the labels of many different trained networks (cf. Liu et al. [2016], Akhtar and Mian [2018]).

A limitation of Theorem 4.1 is that it holds only for datasets of size $m = \mathcal{O}\left(\frac{d}{\max_{i\neq j}|\langle \mathbf{x}_i,\mathbf{x}_j\rangle|}\right)$. E.g., as we discussed in Example 1, if the data points are orthogonal then we need $m = \mathcal{O}(d)$, and if they are random then we need $m = \tilde{\mathcal{O}}(\sqrt{d})$. Moreover, the conditions of the theorem do not allow datasets that contain clusters, where the inner products between data points are large, and do not allow data points which are not of norm $\sqrt{d}$. In the following corollary we extend Theorem 4.1 to allow for such scenarios. Here, the technical assumptions are only on the subset of data points that attain the margin. In particular, if this subset satisfies the assumptions, then the dataset may be arbitrarily large and may contain clusters and points with different norms.

**Corollary 4.1.** *Let $\{(\mathbf{x}_i, y_i)\}_{i=1}^n \subseteq \mathbb{R}^d \times \{-1, 1\}$ be a training dataset. Let $\mathcal{N}_{\boldsymbol{\theta}}$ be a depth-$2$ ReLU network such that $\boldsymbol{\theta}$ is a KKT point of Problem (2). Let $I := \{i \in [n] : y_i\mathcal{N}_{\boldsymbol{\theta}}(\mathbf{x}_i) = 1\}$, $I^+ := \{i \in I : y_i = 1\}$ and $I^- := \{i \in I : y_i = -1\}$. Assume that for all $i \in I$ we have $\|\mathbf{x}_i\| = \sqrt{d}$. Let $m := |I|$, and assume that $\min\left\{\frac{|I^+|}{m}, \frac{|I^-|}{m}\right\} \geq c$ for some $c > 0$, and that $m \leq \frac{d+1}{3(\max_{i\neq j\in I}|\langle \mathbf{x}_i,\mathbf{x}_j\rangle|+1)}$. Then, there is a vector $\mathbf{z} = \eta \cdot \sum_{i\in I} y_i\mathbf{x}_i$ with $\eta > 0$ and $\|\mathbf{z}\| = \mathcal{O}\left(\sqrt{\frac{d}{c^2 m}}\right)$, such that for every $i \in I^+$ we have $\mathcal{N}_{\boldsymbol{\theta}}(\mathbf{x}_i - \mathbf{z}) \leq -1$, and for every $i \in I^-$ we have $\mathcal{N}_{\boldsymbol{\theta}}(\mathbf{x}_i + \mathbf{z}) \geq 1$.*

The proof of the corollary can be easily obtained by slightly modifying the proof of Theorem 4.1 (see Appendix E for details). Note that in Theorem 4.1 the set $I$ of size $m$ contains all the examples in the dataset, and hence for all points in the dataset there are adversarial perturbations of size $\mathcal{O}\left(\sqrt{\frac{d}{c^2 m}}\right)$, while in Corollary 4.1 the set $I$ contains only the examples that attain exactly margin 1, the adversarial perturbations provably exist for examples in $I$, and their size $\mathcal{O}\left(\sqrt{\frac{d}{c^2 m}}\right)$ depends on the size $m$ of $I$.

## 5 Proof sketch of Theorem 4.1

We now discuss the main ideas in the proof of Theorem 4.1. For the formal proof see Appendix D.

### 5.1 A simple example

We start with a simple example to gain some intuition. Consider a dataset $\{(\mathbf{x}_i, y_i)\}_{i=1}^d$ such that for all $i \in [d]$ we have $\mathbf{x}_i = \sqrt{d} \cdot \mathbf{e}_i$, where $\mathbf{e}_1, \ldots, \mathbf{e}_d$ are the standard unit vectors in $\mathbb{R}^d$ and $d$ is even. Suppose that $y_i = 1$ for $i \leq \frac{d}{2}$ and $y_i = -1$ for $i > \frac{d}{2}$.

First, consider the robust network $\mathcal{N}$ of width $d$ from Theorem 3.1 that correctly classifies the dataset. In the proof of Theorem 3.1 we constructed the network $\mathcal{N}(\mathbf{x}) = \sum_{j=1}^d v_j\sigma(\mathbf{w}_j^\top\mathbf{x} + b_j)$ such that for every $j \in [d]$ we have $v_j = y_j$, $\mathbf{w}_j = \frac{2\mathbf{x}_j}{d}$ and $b_j = -1$. Note that we have $y_i\mathcal{N}(\mathbf{x}_i) = 1$ for all $i \in [d]$. In this network, each input $\mathbf{x}_i$ is in the active region (i.e., the region of inputs where the ReLU is active) of exactly one neuron, and has distance of $\frac{\sqrt{d}}{2}$ from the active regions of the other neurons. Hence, adding a perturbation smaller than $\frac{\sqrt{d}}{2}$ to an input $\mathbf{x}_i$ can affect only the contribution of one neuron to the output, and will not flip the output's sign.

Now, we consider a network $\mathcal{N}'(\mathbf{x}) = \sum_{j=1}^d v_j'\sigma(\mathbf{w}_j'^\top\mathbf{x} + b_j')$, such that for all $j \in [d]$ we have $v_j' = y_j$, $\mathbf{w}_j' = \frac{\mathbf{x}_j}{d}$ and $b_j' = 0$. Thus, the weights $\mathbf{w}_j'$ are in the same directions as the weights $\mathbf{w}_j$ of the network $\mathcal{N}$, and in the network $\mathcal{N}'$ the bias terms equal 0. It is easy to verify that for all $i \in [d]$ we have $y_i\mathcal{N}'(\mathbf{x}_i) = 1$. Since $\|\mathbf{w}_j'\| < \|\mathbf{w}_j\|$ and $|b_j'| < |b_j|$ for all $j \in [d]$, then the network $\mathcal{N}'$ is better than $\mathcal{N}$ in the sense of margin maximization. However, $\mathcal{N}'$ is much less robust than $\mathcal{N}$. Indeed, note that in the network $\mathcal{N}'$ each input $\mathbf{x}_i$ is on the boundary of the active regions of all neurons $j \neq i$, that is, for all $j \neq i$ we have $\mathbf{w}_j'^\top\mathbf{x}_i + b_j' = 0$. As a result, a perturbation can affect the contribution of all neurons to the output. Let $i \leq \frac{d}{2}$ and consider adding to $\mathbf{x}_i$ the perturbation $\mathbf{z} = \frac{4}{d}\sum_{j=\frac{d}{2}+1}^d \mathbf{x}_j$. Thus, $\mathbf{z}$ is spanned by all the inputs $\mathbf{x}_j$ where $y_j = -1$, and affects the (negative) contribution of the corresponding neurons. It is not hard to show that $\|\mathbf{z}\| = 2\sqrt{2}$ and $\mathcal{N}'(\mathbf{x}_i + \mathbf{z}) = -1$. Therefore, $\mathcal{N}'$

is much less robust than $\mathcal{N}$ (which required perturbations of size $\frac{\sqrt{d}}{2}$). Thus, bias towards margin maximization might have a negative effect on the robustness. Of course, this is just an example, and in the following subsection we provide a more formal overview of the proof.

## 5.2 Proof overview

We denote $\mathcal{N}_{\boldsymbol{\theta}}(\mathbf{x}) = \sum_{j \in [k]} v_j \sigma(\mathbf{w}_j^\top \mathbf{x} + b_j)$. Thus, $\mathcal{N}_{\boldsymbol{\theta}}$ is a network of width $k$, where the weights in the first layer are $\mathbf{w}_1, \ldots, \mathbf{w}_k$, the bias terms are $b_1, \ldots, b_k$, and the weights in the second layer are $v_1, \ldots, v_k$. We assume that $\mathcal{N}_{\boldsymbol{\theta}}$ satisfies the KKT conditions of Problem (2). We denote $J := [k]$, $J^+ := \{j \in J : v_j \geq 0\}$, and $J^- := \{j \in J : v_j < 0\}$. Note that since the dataset contains both examples with label $1$ and examples with label $-1$ then $J^+$ and $J^-$ are non-empty. For simplicity, we assume that $|v_j| = 1$ for all $j \in J$. That is, $v_j = 1$ for $j \in J^+$ and $v_j = -1$ for $j \in J^-$. We emphasize that we focus here on the case where $|v_j| = 1$ in order to simplify the description of the proof idea, and in the formal proof we do not have such an assumption. We denote $p := \max_{i \neq j} |\langle \mathbf{x}_i, \mathbf{x}_j \rangle|$. Thus, by our assumption we have $m \leq \frac{d+1}{3(p+1)}$.

Since $\boldsymbol{\theta}$ satisfies the KKT conditions (see Appendix A for the formal definition) of Problem (2), then there are $\lambda_1, \ldots, \lambda_m$ such that for every $j \in J$ we have

$$\mathbf{w}_j = \sum_{i \in I} \lambda_i \nabla_{\mathbf{w}_j} \left( y_i \mathcal{N}_{\boldsymbol{\theta}}(\mathbf{x}_i) \right) = \sum_{i \in I} \lambda_i y_i v_j \sigma'_{i,j} \mathbf{x}_i \ , \tag{3}$$

where $\sigma'_{i,j}$ is a subgradient of $\sigma$ at $\mathbf{w}_j^\top \mathbf{x}_i + b_j$, i.e., if $\mathbf{w}_j^\top \mathbf{x}_i + b_j \neq 0$ then $\sigma'_{i,j} = \text{sign}(\mathbf{w}_j^\top \mathbf{x}_i + b_j)$, and otherwise $\sigma'_{i,j}$ is some value in $[0, 1]$ (we note that in this case $\sigma'_{i,j}$ can be any value in $[0, 1]$ and in our proof we do not have any further assumptions on it). Also we have $\lambda_i \geq 0$ for all $i$, and $\lambda_i = 0$ if $y_i \mathcal{N}_{\boldsymbol{\theta}}(\mathbf{x}_i) \neq 1$. Likewise, we have

$$b_j = \sum_{i \in I} \lambda_i \nabla_{b_j} \left( y_i \mathcal{N}_{\boldsymbol{\theta}}(\mathbf{x}_i) \right) = \sum_{i \in I} \lambda_i y_i v_j \sigma'_{i,j} \ . \tag{4}$$

In the proof we use (3) and (4) in order to show that $\mathcal{N}_{\boldsymbol{\theta}}$ is non-robust. We focus here on the case where $i \in I^+$ and we show that $\mathcal{N}_{\boldsymbol{\theta}}(\mathbf{x}_i - \mathbf{z}) \leq -1$. The result for $i \in I^-$ can be obtained in a similar manner. We denote $\mathbf{x}_i' = \mathbf{x}_i - \mathbf{z}$. The proof consists of three main components:

1. We show that $y_i \mathcal{N}_{\boldsymbol{\theta}}(\mathbf{x}_i) = 1$, namely, $\mathbf{x}_i$ attains exactly margin 1.
2. For every $j \in J^+$ we have $\mathbf{w}_j^\top \mathbf{x}_i' + b_j \leq \mathbf{w}_j^\top \mathbf{x}_i + b_j$. Since for $j \in J^+$ we have $v_j = 1$, it implies that when moving from $\mathbf{x}_i$ to $\mathbf{x}_i'$ the non-negative contribution of the neurons in $J^+$ to the output does not increase.
3. When moving from $\mathbf{x}_i$ to $\mathbf{x}_i'$ the total contribution of the neurons in $J^-$ to the output (which is non-positive) decreases by at least 2.

Note that the combination of the above properties imply that $\mathcal{N}_{\boldsymbol{\theta}}(\mathbf{x}_i') \leq -1$ as required. In Appendix C we provide some more details on the main ideas for the proof of each part.

## 6 Experiments

We complement our theoretical results by an empirical study on the robustness of depth-2 ReLU networks trained on synthetically generated datasets. Theorem 4.1 shows that networks trained with gradient flow converge to non-robust networks, but there are still a couple of questions remaining regarding the scope and limitations of this result. First, although the theorem limits the number of samples, we show here that the result applies also in cases when there are much more training samples. Second, the theorem does not depend on the width of the trained network, and we show that even when the size of the training set is much larger than the input dimension, the width of the network does not affect the size of the minimal perturbation that changes the label of the samples.

**Experimental setting.** In all of our experiments we trained a depth-2 fully-connected neural network with ReLU activations using SGD with a batch size of $5,000$. For experiments with less than $5,000$ samples, this is equivalent to full batch gradient descent. We used the exponential loss, although we also tested on logistic loss and obtained similar results. Each experiment was done using

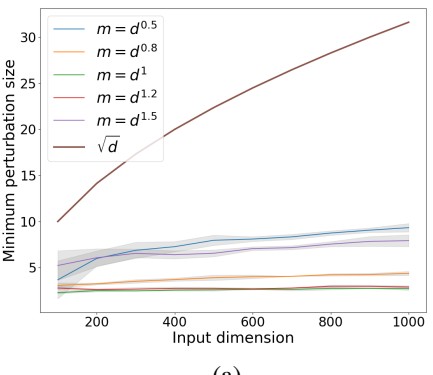
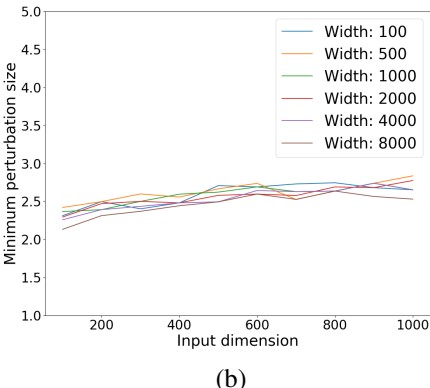

|       (a)       |       (b)       |

Figure 1: The effects of the width and the number of samples on the minimal perturbation size. The x-axis corresponds to the input dimension. The y-axis corresponds to the minimal perturbation size to change the labels of *all* the samples on the margin. We defined the perturbation direction as in Theorem 4.1: $\mathbf{z} := \sum_{i \in I} y_i \mathbf{x}_i$, where $I$ is the set of samples that are on the margin. (a) The minimal perturbation size plotted for different sample sizes. Note that in all of the experiments, the minimal perturbation size is well beyond the plot of $\sqrt{d}$, for which the perturbation is not adversarial. (b) The minimal perturbation size for different widths of the network. Here, the number of samples is $m = d$.

5 different random seeds, and we present the results in terms of the average and (when relevant) standard deviation over these runs. We used an increasing learning rate to accelerate the convergence to the KKT point (which theoretically is only reached at infinity). We began training with a learning rate of $10^{-5}$ and increased it by a factor of $1.1$ every $100$ iterations. We finished training after we achieved a loss smaller than $10^{-30}$. We emphasize that since we use an exponentially tailed loss, the gradients are extremely small at late stages of training, hence to achieve such small loss we must use an increasing learning rate. We implemented our experiments using PyTorch (Paszke et al. [2019]).

**Dataset.** In all of our experiments we sampled $(\mathbf{x}, y) \in \mathbb{R}^d \times \{-1, 1\}$ where $\mathbf{x} \sim \mathcal{U}(\sqrt{d} \cdot \mathbb{S}^{d-1})$ and $y$ is uniform on $\{-1, 1\}$. We also tested on $\mathbf{x}$ sampled from a Gaussian distribution with variance $\frac{1}{d}$ and obtained similar results. Here we only report the results on the uniform distribution.

**Margin.** In our experimental results we defined the margin in the following way: We train a network $\mathcal{N}(\mathbf{x})$ over a dataset $(\mathbf{x}_1, y_1), ..., (\mathbf{x}_m, y_m) \in \mathbb{R}^d \times \{-1, 1\}$. Suppose that after training all the samples are classified correctly (this happened in all of our experiments), i.e. $\mathcal{N}(\mathbf{x}_i) y_i > 0$. We define $i_{\mathrm{marg}} := \mathrm{argmin}_i \mathcal{N}(\mathbf{x}_i) y_i$. Finally, we say that a sample $(\mathbf{x}_i, y_i)$ is *on the margin* if $\mathcal{N}(\mathbf{x}_i) y_i \leq 1.1 \cdot \mathcal{N}(\mathbf{x}_{i_{\mathrm{marg}}}) y_{i_{\mathrm{marg}}}$. In words, we consider $\mathbf{x}_{i_{\mathrm{marg}}}$ to be a sample which is exactly on the margin, but we also allow $10\%$ slack for other samples to be on the margin. We must allow some slack, because in practice we cannot converge exactly to the KKT point, where all the samples on the margin have the exact same output.

### 6.1 Results

**Minimum perturbation size.** Figure 1(a) shows that the perturbation $\mathbf{z}$ defined in Theorem 4.1 can change the labels of all the samples on the margin, even when there are much more samples than stated in Theorem 4.1. To this end, we trained our model on $m = d^\alpha$ samples, where $\alpha = 0.5, 0.8, 1, 1.2, 1.5$ and $d \in \{100, 200, \ldots, 1000\}$. Note that Theorem 4.1 only considers the case of $\alpha = 0.5$ for data which is uniformly distributed. The width of the network is $1,000$. After training is completed, we considered perturbations in the direction of $\mathbf{z} := \sum_{i \in I} y_i \mathbf{x}_i$, where $I$ represents the set of samples that are on the margin. The y-axis represents the minimal $c > 0$ such that for all $i \in I$ we have that $\mathcal{N}\left(\mathbf{x}_i - y_i c \frac{\mathbf{z}}{\|\mathbf{z}\|}\right) \cdot y_i < 0$. In words, we plot the minimal size of the perturbation which changes the labels of *all* the samples on the margin. We emphasize that we used the same perturbation for all the samples.

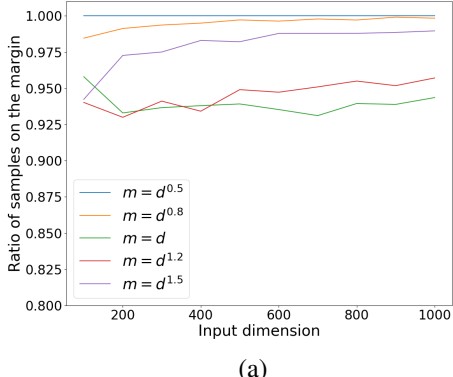 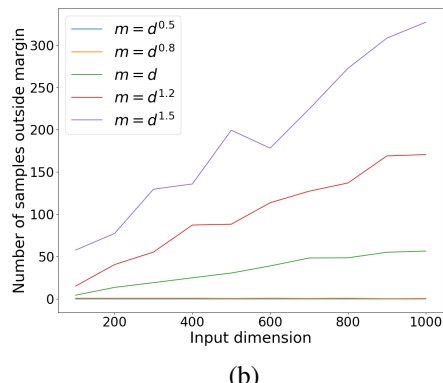

(a)                                             (b)

Figure 2: The number of samples on the margin. The x-axis is the input dimension. Each line plot represents a different number of samples, scaling with the input dimension. (a) The ratio of the number of samples on the margin out of the total number of samples. (b) The number of samples not on the margin.

We also plot $\sqrt{d}$, as a perturbation above this line can trivially change the labels of all points. Recall that by Theorem 3.1, there exists a $\sqrt{d}$-robust network (if the width of the network is at least the size of the dataset). From Figure 1(a), it is clear that the minimal perturbation size is much smaller than $\sqrt{d}$. We also plot the standard deviation over the $5$ different random seeds, showing that our results are consistent.

It is also important to understand how many samples lie on the margin, since our perturbation changes the label of these samples. Figure 2(a) plots the ratio of samples on the margin out of the total number of samples, and Figure 2(b) plots the number of samples not on the margin. These plots correspond to the same experiments as in Figure 1(a), where the number of samples depends on the input dimension. For $m = d^{0.5}$ all the samples are on the margin, as was proven in Lemma D.1. For $m = d^{\alpha}$ where $\alpha = 0.8, 1, 1.2, 1.5$, it can be seen that at least $92\%$ of the samples are on the margin. Together with Figure 1(a), it shows that a single perturbation with small magnitude can change the label of almost all the samples. We remind that this happens when we sample from the uniform distribution, and not when there is a cluster structure as discussed before Corollary 4.1.

**Effect of the width.**  Figure 1(b) shows that the width of the network does not have a significant effect on its robustness. The y-axis is the same as in Figure 1(a), and in all the experiments the number of samples is equal to the input dimension (i.e. $m = d$). We tested on neural networks with width varying from 100 to 8000. The minimal perturbation size in all the experiments is almost the same, regardless of the width. This finding matches the result from Theorem 4.1, but for datasets much larger than the bound from our theory.

## 7  Discussion

In this paper, we showed that in depth-2 ReLU networks gradient flow is biased towards non-robust solutions even when robust solutions exist. To that end, we utilized prior results on the implicit bias of homogeneous models with exponentially-tailed losses. While the existing works on implicit bias are mainly in the context of generalization, in this work we make a first step towards understanding its implications on robustness. We believe that the phenomenon of adversarial examples is an implication of the implicit bias in neural networks, and hence the relationship between these two topics should be further studied. Thus, understanding the implicit bias may be key to explaining the existence of adversarial examples.

We note that we show non-robustness w.r.t. the training dataset rather than on test data. Intuitively, achieving robustness on the training set should be easier than on test data, and we show a negative result already for the former. We believe that extending our approach to robustness on test data is an interesting direction for future research.

There are some additional important open questions that naturally arise from our results. First, our theoretical negative results assume that the size of the dataset (or the size of the subset of examples that attain the margin) is upper bounded by $\mathcal{O}\left(\frac{d}{\max_{i \neq j} |\langle \mathbf{x}_i, \mathbf{x}_j \rangle|}\right)$. This assumption is required in our proof for technical reasons, but we conjecture that it can be significantly relaxed, and our experiments support this conjecture. Another natural question is to extend our results to more architectures, such as deeper ReLU networks. Finally, it would be interesting to study whether other optimization methods have different implicit bias which directs toward robust networks.

## Acknowledgments and Disclosure of Funding

This research is supported in part by European Research Council (ERC) grant 754705.

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
