## A  Preliminaries on the KKT conditions

Below we review the definition of the KKT conditions for non-smooth optimization problems (cf. Lyu and Li [2019], Dutta et al. [2013]).

Let $f : \mathbb{R}^d \to \mathbb{R}$ be a locally Lipschitz function. The Clarke subdifferential [Clarke et al., 2008] at $\mathbf{x} \in \mathbb{R}^d$ is the convex set

$$\partial^\circ f(\mathbf{x}) := \mathrm{conv}\left\{ \lim_{i \to \infty} \nabla f(\mathbf{x}_i) \;\middle|\; \lim_{i \to \infty} \mathbf{x}_i = \mathbf{x},\; f \text{ is differentiable at } \mathbf{x}_i \right\} .$$

If $f$ is continuously differentiable at $\mathbf{x}$ then $\partial^\circ f(\mathbf{x}) = \{\nabla f(\mathbf{x})\}$. For the Clarke subdifferential the chain rule holds as an inclusion rather than an equation. That is, for locally Lipschitz functions $z_1, \ldots, z_n : \mathbb{R}^d \to \mathbb{R}$ and $f : \mathbb{R}^n \to \mathbb{R}$, we have

$$\partial^\circ (f \circ \mathbf{z})(\mathbf{x}) \subseteq \mathrm{conv}\left\{ \sum_{i=1}^n \alpha_i \mathbf{h}_i : \boldsymbol{\alpha} \in \partial^\circ f(z_1(\mathbf{x}), \ldots, z_n(\mathbf{x})), \mathbf{h}_i \in \partial^\circ z_i(\mathbf{x}) \right\} .$$

Consider the following optimization problem

$$\min f(\mathbf{x}) \quad \text{s.t.} \quad \forall n \in [N] \; g_n(\mathbf{x}) \leq 0 , \tag{5}$$

where $f, g_1, \ldots, g_n : \mathbb{R}^d \to \mathbb{R}$ are locally Lipschitz functions. We say that $\mathbf{x} \in \mathbb{R}^d$ is a *feasible point* of Problem (5) if $\mathbf{x}$ satisfies $g_n(\mathbf{x}) \leq 0$ for all $n \in [N]$. We say that a feasible point $\mathbf{x}$ is a *KKT point* if there exists $\lambda_1, \ldots, \lambda_N \geq 0$ such that

1. $\mathbf{0} \in \partial^\circ f(\mathbf{x}) + \sum_{n \in [N]} \lambda_n \partial^\circ g_n(\mathbf{x})$;
2. For all $n \in [N]$ we have $\lambda_n g_n(\mathbf{x}) = 0$.

## B  Proof of Lemma 3.1

Let $\mathbf{x}, \mathbf{x}' \sim \mathcal{U}(\mathbb{S}^{d-1})$ be i.i.d. random variables. Since $\mathbf{x}$ and $\mathbf{x}'$ are independent and uniformly distributed on the sphere, then the distribution of $\mathbf{x}^\top \mathbf{x}'$ equals to the distribution of $\langle \mathbf{x}, (1, 0, \ldots, 0) \rangle$ (i.e., we can assume w.l.o.g. that $\mathbf{x}' = (1, 0, \ldots, 0)^\top$), which equals to the marginal distribution of the first component of $\mathbf{x}$. Let $z$ be the first component of $\mathbf{x}$. By standard results (cf. Fang [2018]), the distribution of $z^2$ is $\mathrm{Beta}(\frac{1}{2}, \frac{d-1}{2})$, namely, a Beta distribution with parameters $\frac{1}{2}, \frac{d-1}{2}$. Thus, the density of $z^2$ is

$$f_{z^2}(y) = \frac{1}{B\left(\frac{1}{2}, \frac{d-1}{2}\right)} y^{-\frac{1}{2}} (1 - y)^{\frac{d-3}{2}} ,$$

where $B(\alpha, \beta) = \frac{\Gamma(\alpha)\Gamma(\beta)}{\Gamma(\alpha+\beta)}$ is the Beta function, and $y \in (0, 1)$. Performing a variable change, we obtain the density of $|z|$, which equals to the density of $|\mathbf{x}^\top \mathbf{x}'|$.

$$f_{|\mathbf{x}^\top \mathbf{x}'|}(y) = f_{|z|}(y) = f_{z^2}(y^2) \cdot 2y = \frac{1}{B\left(\frac{1}{2}, \frac{d-1}{2}\right)} y^{-1}(1 - y^2)^{\frac{d-3}{2}} \cdot 2y = \frac{2}{B\left(\frac{1}{2}, \frac{d-1}{2}\right)} (1 - y^2)^{\frac{d-3}{2}} , \tag{6}$$

where $y \in (0, 1)$. Note that

$$B\left(\frac{1}{2}, \frac{d-1}{2}\right) = \frac{\Gamma(\frac{1}{2})\Gamma(\frac{d-1}{2})}{\Gamma(\frac{d}{2})} \geq \frac{\Gamma(\frac{1}{2})\Gamma(\frac{d}{2}-1)}{\Gamma(\frac{d}{2})} = \frac{\Gamma(\frac{1}{2})}{\frac{d}{2}-1} \geq \frac{2\Gamma(\frac{1}{2})}{d} = \frac{2\sqrt{\pi}}{d} \geq \frac{2}{d} .$$

Combining the above with (6), we obtain

$$f_{|\mathbf{x}^\top \mathbf{x}'|}(y) \leq d(1 - y^2)^{\frac{d-3}{2}} .$$

Therefore, for every $\frac{1}{2} \leq y < 1$ we have $f_{|\mathbf{x}^\top \mathbf{x}'|}(y) \leq d\left(\frac{3}{4}\right)^{\frac{d-3}{2}}$. Hence, we conclude that $\Pr\left[|\langle \mathbf{x}, \mathbf{x}' \rangle| > \frac{1}{2}\right] \leq d\left(\frac{3}{4}\right)^{\frac{d-3}{2}} \cdot 1$. By the union bound, the probability that there are $i \neq j$ such that $|\langle \mathbf{x}_i, \mathbf{x}_j \rangle| > \frac{1}{2}$ is at most

$$m^2 \cdot d \cdot \left(\frac{3}{4}\right)^{\frac{d-3}{2}} \leq d^{2k+1} \left(\frac{3}{4}\right)^{\frac{d-3}{2}} = o_d(1) .$$

Moreover, for every $\frac{\log(d)}{\sqrt{d}} \le y < 1$ we have (for $d \ge 6$)

$$f_{|\mathbf{x}^\top \mathbf{x}'|}(y) \le d(1 - y^2)^{\frac{d-3}{2}} \le d \exp\left(-y^2 \cdot \frac{d-3}{2}\right) \le d \exp\left(-\frac{\log^2(d)}{d} \cdot \frac{d-3}{2}\right)$$

$$\le d \exp\left(-\frac{\ln^2(d)}{d} \cdot \frac{d}{4}\right) = d \cdot d^{-\ln(d)/4} \ .$$

Hence, $\Pr\left[|\langle \mathbf{x}, \mathbf{x}'\rangle| > \frac{\log(d)}{\sqrt{d}}\right] \le d \cdot d^{-\ln(d)/4} \cdot 1$. By the union bound, the probability that there are $i \ne j$ such that $|\langle \mathbf{x}_i, \mathbf{x}_j\rangle| > \frac{\log(d)}{\sqrt{d}}$ is at most

$$m^2 \cdot d \cdot d^{-\ln(d)/4} \le d^{2k+1-\ln(d)/4} = o_d(1) \ .$$

## C  Additional details on the proof sketch for Theorem 4.1

### C.1  Part I: The examples in the dataset attain margin $1$

We show that all examples in the dataset attain margin $1$. The main idea can be described informally as follows (see Lemma D.1 for the details). Assume that there is $i \in I$ such that $y_i \mathcal{N}_{\boldsymbol{\theta}}(\mathbf{x}_i) > 1$. Hence, $\lambda_i = 0$. Suppose w.l.o.g. that $i \in I^+$. Using (3) and (4) we prove that in order to achieve $\mathcal{N}_{\boldsymbol{\theta}}(\mathbf{x}_i) > 1$ when $\lambda_i = 0$, there must be some $r \in I^+$ such that

$$\sum_{j \in J^+} \lambda_r \sigma'_{r,j} = \max_{l \in I}\left(\sum_{j \in J^+} \lambda_l \sigma'_{l,j}\right) > \frac{3}{d+1} \ .$$

Recall that by (3), for $j \in J^+$ the term $\lambda_r \sigma'_{r,j} = \lambda_r v_j \sigma'_{r,j}$ is the coefficient of $y_r \mathbf{x}_r = \mathbf{x}_r$ in the expression for $\mathbf{w}_j$. Hence, $\sum_{j \in J^+} \lambda_r \sigma'_{r,j}$ corresponds to the total sum of coefficients of $\mathbf{x}_r$ over all neurons in $J^+$. Thus, our lower bound on $\sum_{j \in J^+} \lambda_r \sigma'_{r,j}$ implies intuitively that the total sum of coefficients of $\mathbf{x}_r$ is large. We use this fact in order to show that $\mathbf{x}_r$ attains margin strictly larger than $1$, which implies $\lambda_r = 0$ in contradiction to our lower bound on $\sum_{j \in J^+} \lambda_r \sigma'_{r,j}$.

### C.2  Part II: The contribution of the neurons $J^+$ to the output does not increase

For $i \in I^+$ and $\mathbf{x}'_i = \mathbf{x}_i - \mathbf{z} = \mathbf{x}_i - \eta \sum_{l \in I} y_l \mathbf{x}_l$ we show that for every $j \in J^+$ we have $\mathbf{w}_j^\top \mathbf{x}'_i + b_j \le \mathbf{w}_j^\top \mathbf{x}_i + b_j$. Using (3) we have

$$\mathbf{w}_j^\top \sum_{l \in I} y_l \mathbf{x}_l = \sum_{q \in I} \lambda_q y_q v_j \sigma'_{q,j} \mathbf{x}_q^\top \sum_{l \in I} y_l \mathbf{x}_l = \sum_{q \in I} \lambda_q \sigma'_{q,j}\left(y_q^2 \mathbf{x}_q^\top \mathbf{x}_q + \sum_{l \in I,\, l \ne q} y_q y_l \mathbf{x}_q^\top \mathbf{x}_l\right)$$

$$\ge \sum_{q \in I} \lambda_q \sigma'_{q,j}\left(d + \sum_{l \in I,\, l \ne q}(-p)\right) \ge \sum_{q \in I} \lambda_q \sigma'_{q,j}(d - mp) \ .$$

Therefore,

$$\mathbf{w}_j^\top \mathbf{x}'_i + b_j = \mathbf{w}_j^\top \mathbf{x}_i + b_j - \eta \mathbf{w}_j^\top \sum_{l \in I} y_l \mathbf{x}_l \le \mathbf{w}_j^\top \mathbf{x}_i + b_j - \eta \sum_{q \in I} \lambda_q \sigma'_{q,j}(d - mp) \ .$$

By our assumption on $m$ it follows easily that $d - mp > 0$, and hence we conclude that $\mathbf{w}_j^\top \mathbf{x}'_i + b_j \le \mathbf{w}_j^\top \mathbf{x}_i + b_j$.

### C.3  Part III: The contribution of the neurons $J^-$ to the output decreases

We show that for $i \in I^+$ and $\mathbf{x}'_i = \mathbf{x}_i - \mathbf{z} = \mathbf{x}_i - \eta \sum_{l \in I} y_l \mathbf{x}_l$, when moving from $\mathbf{x}_i$ to $\mathbf{x}'_i$ the total contribution of the neurons in $J^-$ to the output decreases by at least $2$. Since for every $j \in J^-$ we

have $v_j = -1$ then we need to show that the sum of the outputs of the neurons $J^-$ increases by at least 2.

By a similar calculation to the one given in Subsection C.2 we obtain that for every $j \in J^-$ and $\eta' > 0$ we have

$$\mathbf{w}_j^\top \left( \mathbf{x}_i - \eta' \sum_{l \in I} y_l \mathbf{x}_l \right) + b_j \geq \mathbf{w}_j^\top \mathbf{x}_i + b_j + \eta' \sum_{q \in I} \lambda_q \sigma'_{q,j} (d - mp) \ . \tag{7}$$

Recall that $d - mp > 0$. Hence, for every $j \in J^-$ the input to neuron $j$ increases by at least $\eta \sum_{q \in I} \lambda_q \sigma'_{q,j} (d - mp)$ when moving from $\mathbf{x}_i$ to $\mathbf{x}'_i$. However, if $\mathbf{w}_j^\top \mathbf{x}_i + b_j < 0$, namely, at $\mathbf{x}_i$ the input to neuron $j$ is negative, then increasing the input may not affect the output of the network. Indeed, by moving from $\mathbf{x}_i$ to $\mathbf{x}'_i$ we might increase the input to neuron $j$ but if it is still negative then the output of neuron $j$ remains 0.

In order to circumvent this issue we analyze the perturbation $\mathbf{z} = \eta \sum_{l \in I} y_l \mathbf{x}_l$ in two stages as follows. We define $\eta = \eta_1 + \eta_2$ for some $\eta_1, \eta_2$ to be chosen later. Let $\tilde{\mathbf{x}}_i = \mathbf{x}_1 - \eta_1 \sum_{l \in I} y_l \mathbf{x}_l$. We prove that for every $j \in J^-$ we have $\mathbf{w}_j^\top \mathbf{x}_i + b_j \geq -(p+1) \sum_{q \in I} \lambda_q \sigma'_{q,j}$, namely, the input to neuron $j$ might be negative but it can be lower bounded. Hence, (7) implies that by choosing $\eta_1 = \frac{p+1}{d-mp}$ we have $\mathbf{w}_j^\top \tilde{\mathbf{x}}_i + b_j \geq 0$ for all $j \in J^-$. That is, in the first stage we move from $\mathbf{x}_i$ to $\tilde{\mathbf{x}}_i$ and increase the inputs to all neurons in $J^-$ such that at $\tilde{\mathbf{x}}_i$ they are least 0. In the second stage we move from $\tilde{\mathbf{x}}_i$ to $\mathbf{x}'_i$ (using the perturbation $\eta_2 \sum_{l \in I} y_l \mathbf{x}_l$). Note that when we move from $\tilde{\mathbf{x}}_i$ to $\mathbf{x}'_i$, every increase in the inputs to the neurons in $J^-$ results in a decrease in the output of the network.

Since we need the output of the network to decrease by at least 2, and since for every $j \in J^-$ we have $v_j = -1$, then when moving from $\tilde{\mathbf{x}}_i$ to $\mathbf{x}'_i$ we need the sum of the inputs to the neurons $J^-$ to increase by at least 2. Similarly to (7), we obtain that when moving from $\tilde{\mathbf{x}}_i$ to $\mathbf{x}'_i$ we increase the sum of the inputs to the neurons $J^-$ by at least

$$\eta_2 \sum_{j \in J^-} \sum_{q \in I} \lambda_q \sigma'_{q,j} (d - mp) \ . \tag{8}$$

Then, we prove a lower bound for $\sum_{j \in J^-} \sum_{q \in I} \lambda_q \sigma'_{q,j}$. We show that such a lower bound can be achieved, since if $\sum_{j \in J^-} \sum_{q \in I} \lambda_q \sigma'_{q,j}$ is too small then it is impossible to have margin 1 for all examples in $I^-$. This lower bound allows us to choose $\eta_2$ such that the expression in (8) is at least 2. Finally, it remains to analyze $\|\mathbf{z}\| = (\eta_1 + \eta_2) \left\| \sum_{l \in I} y_l \mathbf{x}_l \right\|$ and show that it satisfies the required upper bound.

# D  Proof of Theorem 4.1

We start with some required definitions. Some of the definitions are also given in Section 5 and we repeat them here for convenience. We denote $\mathcal{N}_\theta(\mathbf{x}) = \sum_{j \in [k]} v_j \sigma(\mathbf{w}_j^\top \mathbf{x} + b_j)$. Thus, $\mathcal{N}_\theta$ is a network of width $k$, where the weights in the first layer are $\mathbf{w}_1, \ldots, \mathbf{w}_k$, the bias terms are $b_1, \ldots, b_k$, and the weights in the second layer are $v_1, \ldots, v_k$. We denote $J := [k]$, $J^+ := \{j \in J : v_j \geq 0\}$, and $J^- := \{j \in J : v_j < 0\}$. Note that since the dataset contains both examples with label 1 and examples with label $-1$ then $J^+$ and $J^-$ are non-empty. We also denote $p := \max_{i \neq j} |\langle \mathbf{x}_i, \mathbf{x}_j \rangle|$. Since $m \leq \frac{d+1}{3(p+1)}$, we let $c' \leq \frac{1}{3}$ be such that $m = c' \cdot \frac{d+1}{p+1}$. Since $\theta$ satisfies the KKT conditions of Problem (2), then there are $\lambda_1, \ldots, \lambda_m$ such that for every $j \in J$ we have

$$\mathbf{w}_j = \sum_{i \in I} \lambda_i \nabla_{\mathbf{w}_j} (y_i \mathcal{N}_\theta(\mathbf{x}_i)) = \sum_{i \in I} \lambda_i y_i v_j \sigma'_{i,j} \mathbf{x}_i \ , \tag{9}$$

where $\sigma'_{i,j}$ is a subgradient of $\sigma$ at $\mathbf{w}_j^\top \mathbf{x}_i + b_j$, i.e., if $\mathbf{w}_j^\top \mathbf{x}_i + b_j \neq 0$ then $\sigma'_{i,j} = \text{sign}(\mathbf{w}_j^\top \mathbf{x}_i + b_j)$, and otherwise $\sigma'_{i,j}$ is some value in $[0, 1]$. Also we have $\lambda_i \geq 0$ for all $i$, and $\lambda_i = 0$ if $y_i \mathcal{N}_\theta(\mathbf{x}_i) \neq 1$. Likewise, we have

$$b_j = \sum_{i \in I} \lambda_i \nabla_{b_j} (y_i \mathcal{N}_\theta(\mathbf{x}_i)) = \sum_{i \in I} \lambda_i y_i v_j \sigma'_{i,j} \ . \tag{10}$$

**Lemma D.1.** *For all $i \in I$ we have $y_i \mathcal{N}_\theta(\mathbf{x}_i) = 1$.*

*Proof.* Assume that there is $i \in I$ such that $y_i \mathcal{N}_{\boldsymbol{\theta}}(\mathbf{x}_i) > 1$. Hence, $\lambda_i = 0$. If $i \in I^+$, then we have

$$1 < y_i \mathcal{N}_{\boldsymbol{\theta}}(\mathbf{x}_i) = 1 \cdot \sum_{j \in J} v_j \sigma(\mathbf{w}_j^\top \mathbf{x}_i + b_j) \leq \sum_{j \in J^+} v_j \sigma(\mathbf{w}_j^\top \mathbf{x}_i + b_j) \leq \sum_{j \in J^+} v_j \left| \mathbf{w}_j^\top \mathbf{x}_i + b_j \right| .$$

By (9) and (10) the above equals

$$\sum_{j \in J^+} v_j \left| \sum_{l \in I} \lambda_l y_l v_j \sigma'_{l,j} \mathbf{x}_l^\top \mathbf{x}_i + \sum_{l \in I} \lambda_l y_l v_j \sigma'_{l,j} \right| \leq \sum_{j \in J^+} v_j \sum_{l \in I \setminus \{i\}} \left| \lambda_l y_l v_j \sigma'_{l,j} (\mathbf{x}_l^\top \mathbf{x}_i + 1) \right|$$

$$\leq \sum_{j \in J^+} v_j \sum_{l \in I \setminus \{i\}} \lambda_l |y_l v_j| \sigma'_{l,j} (p+1)$$

$$= \sum_{j \in J^+} \sum_{l \in I \setminus \{i\}} v_j^2 \lambda_l \sigma'_{l,j} (p+1)$$

$$= (p+1) \sum_{l \in I \setminus \{i\}} \sum_{j \in J^+} v_j^2 \lambda_l \sigma'_{l,j}$$

$$\leq (p+1) \cdot |I| \cdot \max_{l \in I} \left( \sum_{j \in J^+} v_j^2 \lambda_l \sigma'_{l,j} \right) ,$$

where the first inequality uses $\lambda_i = 0$. Therefore, we have

$$\alpha^+ := \max_{l \in I} \left( \sum_{j \in J^+} v_j^2 \lambda_l \sigma'_{l,j} \right) > \frac{1}{m(p+1)} .$$

From similar arguments, if $i \in I^-$, then we have

$$\alpha^- := \max_{l \in I} \left( \sum_{j \in J^-} v_j^2 \lambda_l \sigma'_{l,j} \right) > \frac{1}{m(p+1)} .$$

Thus, we must have $\max\{\alpha^+, \alpha^-\} > \frac{1}{m(p+1)}$. Assume w.l.o.g. that $\alpha^+ \geq \alpha^-$ (the proof for the case $\alpha^+ < \alpha^-$ is similar). Let $\alpha := \alpha^+$ and $r := \mathrm{argmax}_{l \in I} \left( \sum_{j \in J^+} v_j^2 \lambda_l \sigma'_{l,j} \right)$. Thus, for every $l \in I$ we have $\alpha \geq \sum_{j \in J^+} v_j^2 \lambda_l \sigma'_{l,j}$, $\alpha \geq \sum_{j \in J^-} v_j^2 \lambda_l \sigma'_{l,j}$, and we have $\alpha > \frac{1}{m(p+1)}$. Moreover, we have $\lambda_r > 0$, since otherwise $\alpha = 0$ in contradiction to $\alpha > \frac{1}{m(p+1)} > 0$. Hence, $y_r \mathcal{N}_{\boldsymbol{\theta}}(\mathbf{x}_r) = 1$.

By (9) and (10) we have

$$\mathbf{w}_j^\top \mathbf{x}_r + b_j = \sum_{i \in I} \lambda_i y_i v_j \sigma'_{i,j} \mathbf{x}_i^\top \mathbf{x}_r + \sum_{i \in I} \lambda_i y_i v_j \sigma'_{i,j}$$

$$= \sum_{i \in I} \lambda_i y_i v_j \sigma'_{i,j} (\mathbf{x}_i^\top \mathbf{x}_r + 1)$$

$$= \left( \sum_{i \in I, \, i \neq r} \lambda_i y_i v_j \sigma'_{i,j} (\mathbf{x}_i^\top \mathbf{x}_r + 1) \right) + \lambda_r y_r v_j \sigma'_{r,j} (\mathbf{x}_r^\top \mathbf{x}_r + 1) . \qquad (11)$$

We consider two cases:

**Case 1:** Assume that $r \in I^-$. Let $j \in J^+$. Note that by the definition of $\sigma'_{r,j}$, if $\sigma'_{r,j} \neq 0$ then $\mathbf{w}_j^\top \mathbf{x}_r + b_j \geq 0$. Hence, if $\sigma'_{r,j} \neq 0$ then by (11) we have

$$0 \leq \mathbf{w}_j^\top \mathbf{x}_r + b_j$$

$$= \left( \sum_{i \in I, \, i \neq r} \lambda_i y_i v_j \sigma'_{i,j} (\mathbf{x}_i^\top \mathbf{x}_r + 1) \right) + \lambda_r y_r v_j \sigma'_{r,j} (\mathbf{x}_r^\top \mathbf{x}_r + 1)$$

$$\leq \left( \sum_{i \in I, \, i \neq r} \lambda_i v_j \sigma'_{i,j} (p+1) \right) - \lambda_r v_j \sigma'_{r,j} (d+1) .$$

Thus
$$\lambda_r v_j \sigma'_{r,j}(d+1) \le \sum_{i \in I,\, i \ne r} \lambda_i v_j \sigma'_{i,j}(p+1) \,.$$

Since the above holds for all $j \in J^+$ then
$$\sum_{j \in J^+} v_j^2 \lambda_r \sigma'_{r,j} \le \sum_{j \in J^+} v_j \cdot \frac{1}{d+1} \cdot \sum_{i \in I,\, i \ne r} \lambda_i v_j \sigma'_{i,j}(p+1)$$
$$= \frac{p+1}{d+1} \cdot \sum_{i \in I,\, i \ne r} \sum_{j \in J^+} v_j^2 \lambda_i \sigma'_{i,j}$$
$$\le \frac{p+1}{d+1} \cdot m \cdot \max_{i \in I} \left( \sum_{j \in J^+} v_j^2 \lambda_i \sigma'_{i,j} \right)$$
$$= \frac{p+1}{d+1} \cdot \frac{c'(d+1)}{p+1} \cdot \max_{i \in I} \left( \sum_{j \in J^+} v_j^2 \lambda_i \sigma'_{i,j} \right)$$
$$\le \frac{1}{3} \cdot \max_{i \in I} \left( \sum_{j \in J^+} v_j^2 \lambda_i \sigma'_{i,j} \right) \,,$$

in contradiction to the choice of $r$.

**Case 2:** Assume that $r \in I^+$. We have
$$1 = y_r \mathcal{N}_{\boldsymbol{\theta}}(\mathbf{x}_r) = 1 \cdot \sum_{j \in J} v_j \sigma(\mathbf{w}_j^\top \mathbf{x}_r + b_j)$$
$$\ge \sum_{j \in J^+} v_j \cdot (\mathbf{w}_j^\top \mathbf{x}_r + b_j) + \sum_{j \in J^-} v_j \sigma(\mathbf{w}_j^\top \mathbf{x}_r + b_j) \,. \tag{12}$$

Note that by (11) we have
$$\sum_{j \in J^+} v_j \cdot (\mathbf{w}_j^\top \mathbf{x}_r + b_j) = \sum_{j \in J^+} \left[ \left( \sum_{i \in I,\, i \ne r} \lambda_i y_i v_j^2 \sigma'_{i,j}(\mathbf{x}_i^\top \mathbf{x}_r + 1) \right) + \lambda_r y_r v_j^2 \sigma'_{r,j}(\mathbf{x}_r^\top \mathbf{x}_r + 1) \right]$$
$$\ge \sum_{j \in J^+} \left[ \left( - \sum_{i \in I,\, i \ne r} \lambda_i v_j^2 \sigma'_{i,j}(p+1) \right) + \lambda_r v_j^2 \sigma'_{r,j}(d+1) \right]$$
$$= \left( -(p+1) \sum_{i \in I,\, i \ne r} \sum_{j \in J^+} \lambda_i v_j^2 \sigma'_{i,j} \right) + \sum_{j \in J^+} \lambda_r v_j^2 \sigma'_{r,j}(d+1)$$
$$\ge -(p+1)m\alpha + (d+1)\alpha \,. \tag{13}$$

Moreover, using (11) again we have
$$\sum_{j \in J^-} v_j \sigma\left(\mathbf{w}_j^\top \mathbf{x}_r + b_j\right) = \sum_{j \in J^-} v_j \sigma\left( \left( \sum_{i \in I,\, i \ne r} \lambda_i y_i v_j \sigma'_{i,j}(\mathbf{x}_i^\top \mathbf{x}_r + 1) \right) + \lambda_r y_r v_j \sigma'_{r,j}(\mathbf{x}_r^\top \mathbf{x}_r + 1) \right)$$
$$\ge \sum_{j \in J^-} v_j \sigma\left( \sum_{i \in I,\, i \ne r} \lambda_i y_i v_j \sigma'_{i,j}(\mathbf{x}_i^\top \mathbf{x}_r + 1) \right)$$
$$\ge \sum_{j \in J^-} v_j \left( \sum_{i \in I,\, i \ne r} \lambda_i |v_j| \sigma'_{i,j}(p+1) \right)$$
$$= -(p+1) \left( \sum_{i \in I,\, i \ne r} \sum_{j \in J^-} \lambda_i v_j^2 \sigma'_{i,j} \right)$$
$$\ge -(p+1)m\alpha \,. \tag{14}$$

Combining (12), (13) and (14), we obtain

$$
\begin{aligned}
1 &= y_r \mathcal{N}_{\boldsymbol{\theta}}(\mathbf{x}_r) \\
&\geq -(p+1)m\alpha + (d+1)\alpha - (p+1)m\alpha \\
&= \alpha\left(d + 1 - 2(p+1)m\right) \\
&= \alpha\left(d + 1 - 2(p+1) \cdot \frac{c'(d+1)}{p+1}\right) \\
&= \alpha(d+1)(1 - 2c') \\
&> \frac{1}{m(p+1)} \cdot (d+1)(1-2c') \\
&= \frac{p+1}{c'(d+1)(p+1)} \cdot (d+1)(1-2c') \\
&= \frac{1 - 2c'}{c'} \geq 1 \ .
\end{aligned}
$$

Thus, we reach a contradiction. $\qquad\square$

**Lemma D.2.** *We have*

$$
\sum_{i \in I} \sum_{j \in J^+} v_j^2 \lambda_i \sigma_{i,j}' \geq \frac{mc}{(cc'+1)(d+1)} \ ,
$$

*and*

$$
\sum_{i \in I} \sum_{j \in J^-} v_j^2 \lambda_i \sigma_{i,j}' \geq \frac{mc}{(cc'+1)(d+1)} \ .
$$

*Proof.* We prove here the first claim. The proof of the second claim is similar. For every $i \in I^+$ we have

$$
1 \leq \mathcal{N}_{\boldsymbol{\theta}}(\mathbf{x}_i) \leq \sum_{j \in J^+} v_j \sigma(\mathbf{w}_j^\top \mathbf{x}_i + b_j) \ .
$$

By plugging-in (9) and (10), the above equals to

$$
\sum_{j \in J^+} v_j \sigma\left(\sum_{l \in I} \lambda_l y_l v_j \sigma_{l,j}' \mathbf{x}_l^\top \mathbf{x}_i + \sum_{l \in I} \lambda_l y_l v_j \sigma_{l,j}'\right)
$$

$$
= \sum_{j \in J^+} v_j \sigma\left(\left(\sum_{l \in I,\ l \neq i} \lambda_l y_l v_j \sigma_{l,j}'(\mathbf{x}_l^\top \mathbf{x}_i + 1)\right) + \lambda_i y_i v_j \sigma_{i,j}'(\mathbf{x}_i^\top \mathbf{x}_i + 1)\right)
$$

$$
\leq \sum_{j \in J^+} v_j \sigma\left(\left(\sum_{l \in I,\ l \neq i} \sigma\left(\lambda_l y_l v_j \sigma_{l,j}'(\mathbf{x}_l^\top \mathbf{x}_i + 1)\right)\right) + \sigma\left(\lambda_i y_i v_j \sigma_{i,j}'(\mathbf{x}_i^\top \mathbf{x}_i + 1)\right)\right)
$$

$$
= \sum_{j \in J^+} v_j \left(\left(\sum_{l \in I,\ l \neq i} \sigma\left(\lambda_l y_l v_j \sigma_{l,j}'(\mathbf{x}_l^\top \mathbf{x}_i + 1)\right)\right) + \lambda_i v_j \sigma_{i,j}'(d+1)\right)
$$

$$
\leq \sum_{j \in J^+} v_j \left(\left(\sum_{l \in I,\ l \neq i} \lambda_l v_j \sigma_{l,j}'(p+1)\right) + \lambda_i v_j \sigma_{i,j}'(d+1)\right)
$$

$$
= \left(\sum_{l \in I,\ l \neq i} \sum_{j \in J^+} \lambda_l v_j^2 \sigma_{l,j}'(p+1)\right) + \sum_{j \in J^+} \lambda_i v_j^2 \sigma_{i,j}'(d+1) \ .
$$

Let $\gamma = \frac{1}{cc'+1}$. By the above equation, for every $i \in I^+$ we either have

$$
(p+1) \sum_{l \in I,\ l \neq i} \sum_{j \in J^+} \lambda_l v_j^2 \sigma_{l,j}' \geq 1 - \gamma \ , \tag{15}
$$

or

$$(d+1)\sum_{j\in J^+}\lambda_i v_j^2 \sigma'_{i,j} \ge \gamma \,. \tag{16}$$

If there exists $i \in I^+$ such that (15) holds, then we have

$$(p+1)\sum_{l\in I}\sum_{j\in J^+}\lambda_l v_j^2\sigma'_{l,j} \ge (p+1)\sum_{l\in I,\, l\ne i}\sum_{j\in J^+}\lambda_l v_j^2\sigma'_{l,j} \ge 1-\gamma = \frac{cc'}{cc'+1}\,,$$

and hence

$$\sum_{l\in I}\sum_{j\in J^+}\lambda_l v_j^2\sigma'_{l,j} \ge \frac{cc'}{(cc'+1)(p+1)} = \frac{m}{d+1}\cdot\frac{c}{cc'+1} = \frac{mc}{(cc'+1)(d+1)}$$

as required. Otherwise, namely, if for every $i \in I^+$ (15) does not hold, then for every $i \in I^+$ (16) holds, and therefore

$$\sum_{i\in I}\sum_{j\in J^+}\lambda_i v_j^2\sigma'_{i,j} \ge \sum_{i\in I^+}\sum_{j\in J^+}\lambda_i v_j^2\sigma'_{i,j} \ge |I^+|\cdot\frac{\gamma}{d+1} \ge \frac{mc}{(cc'+1)(d+1)}\,.$$

$\square$

**Lemma D.3.** *Let $i \in I$ and $j \in J$. Then,*

$$\mathbf{w}_j^\top \mathbf{x}_i + b_j \ge -(p+1)\sum_{l\in I}|v_j|\lambda_l\sigma'_{l,j}\,.$$

*Proof.* If $\mathbf{w}_j^\top \mathbf{x}_i + b_j \ge 0$ then the claim follows immediately. Otherwise, $\sigma'_{i,j} = 0$. Hence, by (9) and (10) we have

$$\begin{aligned}
\mathbf{w}_j^\top \mathbf{x}_i + b_j &= \sum_{l\in I}\lambda_l y_l v_j \sigma'_{l,j}\mathbf{x}_l^\top \mathbf{x}_i + \sum_{l\in I}\lambda_l y_l v_j\sigma'_{l,j}\\
&= \sum_{l\in I,\, l\ne i}\lambda_l y_l v_j\sigma'_{l,j}(\mathbf{x}_l^\top \mathbf{x}_i + 1)\\
&\ge -(p+1)\sum_{l\in I,\, l\ne i}\lambda_l|v_j|\sigma'_{l,j}\\
&= -(p+1)\sum_{l\in I}\lambda_l|v_j|\sigma'_{l,j}\,.
\end{aligned}$$

$\square$

**Lemma D.4.** *Let $\mathbf{u} = \sum_{l\in I} y_l\mathbf{x}_l$. For every $j \in J^+$ we have $\mathbf{w}_j^\top\mathbf{u} \ge \sum_{i\in I}v_j\lambda_i\sigma'_{i,j}(d-mp)$. For every $j \in J^-$ we have $\mathbf{w}_j^\top\mathbf{u} \le \sum_{i\in I}v_j\lambda_i\sigma'_{i,j}(d-mp)$.*

*Proof.* For $j \in J^+$, using (9) we have

$$\begin{aligned}
\mathbf{w}_j^\top\sum_{l\in I}y_l\mathbf{x}_l &= \sum_{i\in I}\lambda_i y_i v_j\sigma'_{i,j}\mathbf{x}_i^\top\sum_{l\in I}y_l\mathbf{x}_l\\
&= \sum_{i\in I}\lambda_i v_j\sigma'_{i,j}\left(y_i^2\mathbf{x}_i^\top\mathbf{x}_i + \sum_{l\in I,\, l\ne i}y_i y_l\mathbf{x}_i^\top\mathbf{x}_l\right)\\
&\ge \sum_{i\in I}\lambda_i v_j\sigma'_{i,j}\left(d + \sum_{l\in I,\, l\ne i}(-p)\right)\\
&\ge \sum_{i\in I}\lambda_i v_j\sigma'_{i,j}(d-mp)\,.
\end{aligned}$$

Likewise, for $j \in J^-$ we have

$$\mathbf{w}_j^\top \sum_{l \neq I} y_l \mathbf{x}_l = \sum_{i \in I} \lambda_i v_j \sigma'_{i,j} \left( y_i^2 \mathbf{x}_i^\top \mathbf{x}_i + \sum_{l \in I, \, l \neq i} y_i y_l \mathbf{x}_i^\top \mathbf{x}_l \right)$$

$$\leq \sum_{i \in I} \lambda_i v_j \sigma'_{i,j} \left( d + \sum_{l \in I, \, l \neq i} (-p) \right)$$

$$\leq \sum_{i \in I} \lambda_i v_j \sigma'_{i,j} (d - mp) \ .$$

$\square$

**Lemma D.5.** *We have $d - mp > 0$.*

*Proof.* By our assumption on $m$ we have

$$d - mp = d - \frac{c'(d+1)p}{p+1} \geq d - \frac{1}{3} \cdot \frac{2dp}{p} = d - \frac{2d}{3} > 0 \ .$$

$\square$

**Lemma D.6.** *Let $\mathbf{z} = \eta \sum_{l \in I} y_l \mathbf{x}_l$ for some $\eta \geq \frac{p+1}{d-mp}$. Let $i \in I$. For all $j \in J^-$ we have $\mathbf{w}_j^\top (\mathbf{x}_i - \mathbf{z}) + b_j \geq 0$, and for all $j \in J^+$ we have $\mathbf{w}_j^\top (\mathbf{x}_i + \mathbf{z}) + b_j \geq 0$.*

*Proof.* Let $j \in J^-$. By Lemma D.3 we have $\mathbf{w}_j^\top \mathbf{x}_i + b_j \geq (p+1) \sum_{l \in I} v_j \lambda_l \sigma'_{l,j}$. By Lemma D.4 we have $-\mathbf{w}_j^\top \mathbf{z} \geq -\eta \sum_{l \in I} v_j \lambda_l \sigma'_{l,j} (d - mp)$. By combining these results we obtain

$$\mathbf{w}_j^\top (\mathbf{x}_i - \mathbf{z}) + b_j = \mathbf{w}_j^\top \mathbf{x}_i + b_j - \mathbf{w}_j^\top \mathbf{z}$$

$$\geq (p+1) \sum_{l \in I} v_j \lambda_l \sigma'_{l,j} - \eta \sum_{l \in I} v_j \lambda_l \sigma'_{l,j} (d - mp)$$

$$= \sum_{l \in I} v_j \lambda_l \sigma'_{l,j} (p + 1 - \eta(d - mp)) \ .$$

Note that by Lemma D.5 we have $d - mp > 0$. Hence, for $\eta \geq \frac{p+1}{d-mp}$ we have $\mathbf{w}_j^\top (\mathbf{x}_i - \mathbf{z}) + b_j \geq 0$.

Let $j \in J^+$. By Lemma D.3 we have $\mathbf{w}_j^\top \mathbf{x}_i + b_j \geq -(p+1) \sum_{l \in I} v_j \lambda_l \sigma'_{l,j}$. By Lemma D.4 we have $\mathbf{w}_j^\top \mathbf{z} \geq \eta \sum_{l \in I} v_j \lambda_l \sigma'_{l,j} (d - mp)$. By combining these results we obtain

$$\mathbf{w}_j^\top (\mathbf{x}_i + \mathbf{z}) + b_j = \mathbf{w}_j^\top \mathbf{x}_i + b_j + \mathbf{w}_j^\top \mathbf{z}$$

$$\geq -(p+1) \sum_{l \in I} v_j \lambda_l \sigma'_{l,j} + \eta \sum_{l \in I} v_j \lambda_l \sigma'_{l,j} (d - mp)$$

$$= \sum_{l \in I} v_j \lambda_l \sigma'_{l,j} (-(p+1) + \eta(d - mp)) \ .$$

Hence, for $\eta \geq \frac{p+1}{d-mp}$ we have $\mathbf{w}_j^\top (\mathbf{x}_i + \mathbf{z}) + b_j \geq 0$. $\square$

Let $\eta_1 = \frac{p+1}{d-mp}$ and $\eta_2 = \frac{2(cc'+1)(d+1)}{mc(d-mp)}$. Note that by Lemma D.5 both $\eta_1$ and $\eta_2$ are positive. We denote $\mathbf{z} = (\eta_1 + \eta_2) \sum_{l \in I} y_l \mathbf{x}_l$.

**Lemma D.7.** *Let $i \in I^+$, and let $\mathbf{x}_i' = \mathbf{x}_i - \mathbf{z}$. Then, $\mathcal{N}_{\boldsymbol{\theta}}(\mathbf{x}_i') \leq -1$.*

*Proof.* By Lemma D.4, for every $j \in J^+$ we have

$$\mathbf{w}_j^\top \mathbf{x}_i' + b_j = \mathbf{w}_j^\top \mathbf{x}_i + b_j - \mathbf{w}_j^\top (\eta_1 + \eta_2) \sum_{l \in I} y_l \mathbf{x}_l \leq \mathbf{w}_j^\top \mathbf{x}_i + b_j - (\eta_1 + \eta_2) \sum_{l \in I} v_j \lambda_l \sigma'_{l,j} (d - mp) \ .$$

By Lemma D.5 the above is at most $\mathbf{w}_j^\top \mathbf{x}_i + b_j$.

Consider now $j \in J^-$. Let $\tilde{\mathbf{x}}_i = \mathbf{x}_i - \eta_1 \sum_{l \in I} y_l \mathbf{x}_l$. By Lemma D.6 we have $\mathbf{w}_j^\top \tilde{\mathbf{x}}_i + b_j \geq 0$. Also, by Lemma D.4 we have

$$\mathbf{w}_j^\top \tilde{\mathbf{x}}_i + b_j = \mathbf{w}_j^\top \mathbf{x}_i + b_j - \mathbf{w}_j^\top \eta_1 \sum_{l \in I} y_l \mathbf{x}_l \geq \mathbf{w}_j^\top \mathbf{x}_i + b_j - \eta_1 \sum_{l \in I} v_j \lambda_l \sigma'_{l,j}(d - mp) \,,$$

and by Lemma D.5 the above is at least $\mathbf{w}_j^\top \mathbf{x}_i + b_j$. Hence,

$$\mathbf{w}_j^\top \mathbf{x}'_i + b_j = \mathbf{w}_j^\top \tilde{\mathbf{x}}_i + b_j - \mathbf{w}_j^\top \eta_2 \sum_{l \in I} y_l \mathbf{x}_l \geq \max\left\{0, \mathbf{w}_j^\top \mathbf{x}_i + b_j\right\} - \eta_2 \cdot \mathbf{w}_j^\top \sum_{l \in I} y_l \mathbf{x}_l \,.$$

By Lemma D.4, the above is at least

$$\max\left\{0, \mathbf{w}_j^\top \mathbf{x}_i + b_j\right\} - \eta_2 \sum_{l \in I} v_j \lambda_l \sigma'_{l,j}(d - mp) \,.$$

Overall, we have

$$\mathcal{N}_{\boldsymbol{\theta}}(\mathbf{x}'_i) = \sum_{j \in J^+} v_j \sigma(\mathbf{w}_j^\top \mathbf{x}'_i + b_j) + \sum_{j \in J^-} v_j \sigma(\mathbf{w}_j^\top \mathbf{x}'_i + b_j)$$

$$\leq \sum_{j \in J^+} v_j \sigma(\mathbf{w}_j^\top \mathbf{x}_i + b_j) + \sum_{j \in J^-} v_j \left(\max\left\{0, \mathbf{w}_j^\top \mathbf{x}_i + b_j\right\} - \eta_2 \sum_{l \in I} v_j \lambda_l \sigma'_{l,j}(d - mp)\right)$$

$$= \sum_{j \in J} v_j \sigma(\mathbf{w}_j^\top \mathbf{x}_i + b_j) - \sum_{j \in J^-} v_j \eta_2 \sum_{l \in I} v_j \lambda_l \sigma'_{l,j}(d - mp)$$

$$= \mathcal{N}_{\boldsymbol{\theta}}(\mathbf{x}_i) - \eta_2(d - mp) \sum_{j \in J^-} \sum_{l \in I} v_j^2 \lambda_l \sigma'_{l,j} \,.$$

By Lemmas D.1, D.2 and D.5 the above is at most

$$1 - \eta_2(d - mp) \cdot \frac{mc}{(cc' + 1)(d + 1)} \,.$$

For $\eta_2 = \frac{2(cc'+1)(d+1)}{mc(d-mp)}$ we conclude that $\mathcal{N}_{\boldsymbol{\theta}}(\mathbf{x}'_i)$ is at most $-1$. $\qquad\square$

**Lemma D.8.** *Let $i \in I^-$, and let $\mathbf{x}'_i = \mathbf{x}_i + \mathbf{z}$. Then, $\mathcal{N}_{\boldsymbol{\theta}}(\mathbf{x}'_i) \geq 1$.*

*Proof.* The proof follows similar arguments to the proof of Lemma D.7. We give it here for completeness.

By Lemma D.4, for every $j \in J^-$ we have

$$\mathbf{w}_j^\top \mathbf{x}'_i + b_j = \mathbf{w}_j^\top \mathbf{x}_i + b_j + \mathbf{w}_j^\top(\eta_1 + \eta_2) \sum_{l \in I} y_l \mathbf{x}_l \leq \mathbf{w}_j^\top \mathbf{x}_i + b_j + (\eta_1 + \eta_2) \sum_{l \in I} v_j \lambda_l \sigma'_{l,j}(d - mp) \,.$$

By Lemma D.5 the above is at most $\mathbf{w}_j^\top \mathbf{x}_i + b_j$.

Consider now $j \in J^+$. Let $\tilde{\mathbf{x}}_i = \mathbf{x}_i + \eta_1 \sum_{l \in I} y_l \mathbf{x}_l$. By Lemma D.6 we have $\mathbf{w}_j^\top \tilde{\mathbf{x}}_i + b_j \geq 0$. Also, by Lemma D.4 we have

$$\mathbf{w}_j^\top \tilde{\mathbf{x}}_i + b_j = \mathbf{w}_j^\top \mathbf{x}_i + b_j + \mathbf{w}_j^\top \eta_1 \sum_{l \in I} y_l \mathbf{x}_l \geq \mathbf{w}_j^\top \mathbf{x}_i + b_j + \eta_1 \sum_{l \in I} v_j \lambda_l \sigma'_{l,j}(d - mp) \,,$$

and by Lemma D.5 the above is at least $\mathbf{w}_j^\top \mathbf{x}_i + b_j$. Hence,

$$\mathbf{w}_j^\top \mathbf{x}'_i + b_j = \mathbf{w}_j^\top \tilde{\mathbf{x}}_i + b_j + \mathbf{w}_j^\top \eta_2 \sum_{l \in I} y_l \mathbf{x}_l \geq \max\left\{0, \mathbf{w}_j^\top \mathbf{x}_i + b_j\right\} + \eta_2 \cdot \mathbf{w}_j^\top \sum_{l \in I} y_l \mathbf{x}_l \,.$$

By Lemma D.4, the above is at least

$$\max\left\{0, \mathbf{w}_j^\top \mathbf{x}_i + b_j\right\} + \eta_2 \sum_{l \in I} v_j \lambda_l \sigma'_{l,j}(d - mp) \,.$$

Overall, we have

$$\mathcal{N}_{\boldsymbol{\theta}}(\mathbf{x}_i') = \sum_{j \in J^+} v_j \sigma(\mathbf{w}_j^\top \mathbf{x}_i' + b_j) + \sum_{j \in J^-} v_j \sigma(\mathbf{w}_j^\top \mathbf{x}_i' + b_j)$$

$$\geq \sum_{j \in J^+} v_j \left( \max\left\{0, \mathbf{w}_j^\top \mathbf{x}_i + b_j\right\} + \eta_2 \sum_{l \in I} v_j \lambda_l \sigma_{l,j}'(d - mp) \right) + \sum_{j \in J^-} v_j \sigma(\mathbf{w}_j^\top \mathbf{x}_i + b_j)$$

$$= \sum_{j \in J} v_j \sigma(\mathbf{w}_j^\top \mathbf{x}_i + b_j) + \sum_{j \in J^+} v_j \eta_2 \sum_{l \in I} v_j \lambda_l \sigma_{l,j}'(d - mp)$$

$$= \mathcal{N}_{\boldsymbol{\theta}}(\mathbf{x}_i) + \eta_2(d - mp) \sum_{j \in J^+} \sum_{l \in I} v_j^2 \lambda_l \sigma_{l,j}' \ .$$

By Lemmas D.1, D.2 and D.5 the above is at least

$$-1 + \eta_2(d - mp) \cdot \frac{mc}{(cc' + 1)(d + 1)} \ .$$

For $\eta_2 = \frac{2(cc'+1)(d+1)}{mc(d-mp)}$ we conclude that $\mathcal{N}_{\boldsymbol{\theta}}(\mathbf{x}_i')$ is at least 1. $\qquad\square$

**Lemma D.9.** *We have* $\|\mathbf{z}\| = \mathcal{O}\left(\sqrt{\frac{d}{c^2 m}}\right)$.

*Proof.* We have

$$\|\mathbf{z}\|^2 = (\eta_1 + \eta_2)^2 \left\| \sum_{l \in I} y_l \mathbf{x}_l \right\|^2 = (\eta_1 + \eta_2)^2 \sum_{l \in I} \sum_{l' \in I} y_l y_{l'} \langle \mathbf{x}_l, \mathbf{x}_{l'} \rangle$$

$$\leq \left( \frac{p+1}{d - mp} + \frac{2(cc'+1)(d+1)}{mc(d-mp)} \right)^2 \left( md + m^2 p \right)$$

$$\leq \left( \frac{p+1}{d - m(p+1)} + \frac{2(cc'+1)(d+1)}{mc(d-m(p+1))} \right)^2 \left( md + m^2(p+1) \right)$$

$$= \left( \frac{p+1}{d - c'(d+1)} + \frac{2(cc'+1)(d+1)}{mc(d-c'(d+1))} \right)^2 \left( md + mc'(d+1) \right)$$

$$\leq \left( \frac{p+1}{\frac{d+1}{2} - \frac{1}{3} \cdot (d+1)} + \frac{2(cc'+1)(d+1)}{mc\left(\frac{d+1}{2} - \frac{1}{3} \cdot (d+1)\right)} \right)^2 \left( md + m \cdot \frac{1}{3} \cdot 2d \right)$$

$$\leq \left( \frac{p+1}{\frac{d+1}{6}} + \frac{2(cc'+1)(d+1)}{mc\left(\frac{d+1}{6}\right)} \right)^2 (2md) = \left( \frac{6c'}{m} + \frac{12(cc'+1)}{mc} \right)^2 (2md)$$

$$= \left( \frac{(18cc'+12)\sqrt{2md}}{mc} \right)^2 \leq \left( \frac{(6+12)\sqrt{2d}}{c\sqrt{m}} \right)^2 \ ,$$

where in the last inequality we used both $c' \leq \frac{1}{3}$ and $c \leq 1$. Hence, $\|\mathbf{z}\| = \mathcal{O}\left(\sqrt{\frac{d}{c^2 m}}\right)$. $\qquad\square$

The theorem now follows immediately form Lemmas D.7, D.8 and D.9.

## E    Proof of Corollary 4.1

The expressions for $\mathbf{w}_j$ and $b_j$ given in (9) and (10) depend only on the examples $(\mathbf{x}_i, y_i)$ where $y_i \mathcal{N}_{\boldsymbol{\theta}}(\mathbf{x}) = 1$. Indeed, if $y_i \mathcal{N}_{\boldsymbol{\theta}}(\mathbf{x}) \neq 1$ then $\lambda_i = 0$. Thus, all examples in the dataset that do not attain margin 1 in $\mathcal{N}_{\boldsymbol{\theta}}$ do not affect the expressions that describe the network $\mathcal{N}_{\boldsymbol{\theta}}$. All arguments in the proof of Theorem 4.1 require only the examples $(\mathbf{x}_i, y_i)$ that appear in (9) and (10). As a consequence, all parts in the proof of Theorem 4.1 hold also here w.r.t. the set $I$. That is, the fact that the dataset includes additional points that do not appear (9) and (10) does not affect the proof. The only part of the proof of Theorem 4.1 that is not required here is Lemma D.1, since we assume that all points in $I$ satisfy $y_i \mathcal{N}_{\boldsymbol{\theta}}(\mathbf{x}_i) = 1$.