# OpenReview forum: "Gradient Methods Provably Converge to Non-Robust Networks"
_NeurIPS.cc/2022/Conference — NeurIPS 2022 Accept_

### Official Review · Reviewer_51Wy · 2022-06-19

**Rating:** 6
**Confidence:** 4
**Soundness:** 3 good
**Presentation:** 3 good
**Contribution:** 3 good

**Summary:**

This paper firstly shows that in depth-2 ReLU networks gradient flow is biased towards non-robust solutions even when robust solutions exist theoretically, and then verifies it experimentally on synthetically generated datasets.

**Questions:**

There is perhaps a minor problem in the statement of Theorem 3.1. The proof uses the condition $m \le k$ but it do not appear in the statement of the theory (Usually, we regard the architectures as fixed before learning, so I think it is better to fix $k$ at first and state that $m \le k$).

**Limitations:**

The paper stated the limitations about the theory, such as the limitations of the training set size, the results are just guaranteed for the training set, and so on.

**Strengths And Weaknesses:**

Strengths:
- This paper shows a novel result that in depth-2 ReLU networks gradient flow is biased towards non-robust solutions even when robust solutions exist theoretically. And then further verifies it in the experiments.
- This paper gives some intuitive examples that help understanding the theorys and ideas to prove the theory.
- This paper expresses its' ideas clearly.
- This paper connects the theoretical results to some empirical phenomenons (line 198-206), although the empirical phenomenons are usually discussed for the test set and the theory in this paper is for the training set.

Weaknesses:
- This main contribution of this paper is just Theorem 4.1 and a naive result Theorem 3.1 and simulation experiments, and the models and datas is somehow simple, I think the amount of work is not so enough in some sense, but the phenomenon it reveals is quite interesting.

---

> ### Author Response · Authors · 2022-07-31
> **Rebuttal**
>
> We thank the reviewer for the feedback and the comments.
>
> “This main contribution of this paper is just Theorem 4.1 and a naive result Theorem 3.1 and simulation experiments, and the models and datas is somehow simple, I think the amount of work is not so enough in some sense, but the phenomenon it reveals is quite interesting.“
>
> As far as we are aware, our paper is the first work that gives theoretical guarantees for the interesting phenomenon that networks trained with standard gradient methods tend to be non-robust. We also think that our approach of analyzing a trained network via the KKT conditions derived from the theoretical results on implicit bias is interesting and may be applied to other problems related to properties of trained neural networks (i.e. properties beyond robustness). Also, the proof of Theorem 4.1 (which we also extend to Corollary 4.1) is non-trivial and requires significant work.
>
> “There is perhaps a minor problem in the statement of Theorem 3.1. The proof uses the condition $m \leq k$ but it does not appear in the statement of the theory (Usually, we regard the architectures as fixed before learning, so I think it is better to fix $k$ at first and state that $m \leq k$).”
>
> The theorem guarantees the existence of a robust network of width $m$, which immediately implies that a robust network exists for every width $k \geq m$. However, we agree that stating the theorem as you suggested might be clearer, and we will rephrase it.

---

### Official Review · Reviewer_GV2f · 2022-06-28

**Rating:** 7
**Confidence:** 3
**Soundness:** 3 good
**Presentation:** 3 good
**Contribution:** 3 good

**Summary:**

This paper studied the adversarial examples from a theoretical perspective. Specifically, the authors considered a depth-2 ReLU homogeneous neural network trained with gradient flow, and then showed that it is non-robust (under $l_2$ adversarial perturbations), even when a robust neural network does exist.

**Questions:**

1. The authors showed that when training data points are neither too many (compared with the dimension) nor too correlated, the gradient flow converges to non-robust solutions in spite of robust ones existing. What if there are a lot of data points and many of them are correlated? Will the result be the same or do adversarial examples only exist for some training data points?

2. In the paper, the authors assumed all the input features have the same norm, however, it may not hold in the real application, for instance, different images have different norms. Thus, I was wondering how this assumption will affect the results.

3. In Figure 2 (a), the purple line with the most data points, why is the ratio of sample on the margin higher than the green and red lines?

**Ethics Review Area:**

["I don’t know"]

**Strengths And Weaknesses:**

Strength:

1. The authors used a depth-2 ReLU neural network to provide a theoretical explanation for the existence of adversarial examples in the trained network.

2. The authors conducted some numerical experiments, and their results supported the theoretical results.

3. The result of the paper is very important, which explains that even with a well-trained neural network, there always exist some adversarial examples to fool it.

Weakness:

The assumption for the theoretical results is quite strong in the big data era, for example, in Theorem 4.1, the data size is even smaller than the dimension; in Corollary 4.1, just curious, can we always get such a network $N_\theta$ with KKT point and the data set $I:=  {i\in [n]: y_iN_{\theta}(x_i) = 1\}$ is non-empty. Maybe the authors should elaborate on this a bit more.

---

> ### Author Response · Authors · 2022-07-31
> **Rebuttal**
>
> We thank the reviewer for the thorough review and the constructive feedback.
>
> “in Corollary 4.1, just curious, can we always get such a network $N_\theta$ with KKT point and the data set $I := i \in [n]$ s.t. $y_iN_\theta(x_i)=1$ is non-empty. Maybe the authors should elaborate on this a bit more.”
>
> If understood correctly, the question is whether it is possible that the set $I$ is empty. This is not possible, because in this case, the complementary slackness condition would imply that $\lambda_i=0$ for all $i$, and then the stationarity condition would imply that $\theta=0$, i.e. the network outputs zero for every input.
>
> To answer the reviewer’s questions:
>
> 1) If there are many data points and many of them are correlated, but the subset of points on the margin is not too large and not too correlated, then Corollary 4.1 would imply the existence of adversarial examples for the points on the margin.
> If the subset of points on the margin is either too large or too correlated then our results do not imply any guarantee for the existence of adversarial examples, and understanding this case may be an intriguing direction for future research. We note that we showed empirically that our result holds also where the size of the dataset is much larger than the input dimension.
>
> 2) The assumption about the points having the same norm is for simplicity. We believe that our result can be extended to the case where the norms are not equal, as long as the ratio between the maximal and minimal norm of points in the dataset is bounded.
>
> 3) This is a good question. This phenomenon occurs empirically, but we do not have a theoretical explanation for it.

---

### Official Review · Reviewer_ysvx · 2022-07-11

**Rating:** 6
**Confidence:** 2
**Soundness:** 3 good
**Presentation:** 2 fair
**Contribution:** 3 good

**Summary:**

The paper theoretically shows that for a 2-layer ReLU network trained by gradient methods, it provably converges to non-robust networks (regards to l2 perturbation) even when a robust network exits. They support their theoretical findings by experiments with a 2-layer fully connected network with ReLU activation on synthetically generated datasets.


**Questions:**

1. Will it be easy to extend the proof to other kinds of robustness, for example,  $L_{\inf}$ robustness?

2. How is the magnitude of the perturbations used in the theoretical analysis compared to those used in the practical settings in Cifar or ImageNet?


**Limitations:**

The authors fairly addressed the limitations and potential negative societal impact of their work.

**Strengths And Weaknesses:**

This paper mainly focuses on the theoretical analysis of why adversarial examples exist. They prove that given a two-way classification problem, a robust 2-layer ReLU network exists, but gradient flow converges to non-robust networks. The paper is interesting to read although I did not read every line of the proof, and the claim that converging to non-robust networks is the implicit bias seems interesting and novel to me. The properties of the theorem are interesting and consistent with the existing findings.

It will be even more interesting if it can be extended to more realistic or practical settings, i.e., more complicated networks and robustness other than l2 distance.

---

> ### Author Response · Authors · 2022-07-31
> **Rebuttal**
>
> We thank the reviewer for the feedback. To answer the reviewer’s questions:
>
> “It will be even more interesting if it can be extended to more realistic or practical settings, i.e., more complicated networks and robustness other than l2 distance.”
>
> We agree. These are intriguing directions for future research.
>
> “Will it be easy to extend the proof to other kinds of robustness, for example, $L_\inf$ robustness?“
>
> This is a good question. It would require a somewhat different setting, but it might be possible. For example, if we want to consider robustness w.r.t. $L_\inf$, then we need to assume that each pair of data points are far w.r.t. this norm.
>
> “How is the magnitude of the perturbations used in the theoretical analysis compared to those used in the practical settings in CIFAR or ImageNet?“
>
> Some typical magnitudes of perturbations can be found in [1]. E.g., perturbation of 0.5 in CIFAR-10.
> However, the numbers in our analysis are not directly comparable, due to differences in the technical assumptions.
>
> [1] RobustBench: a standardized adversarial robustness benchmark, Croce, Francesco and Andriushchenko, Maksym and Sehwag, Vikash and Debenedetti, Edoardo and Flammarion, Nicolas and Chiang, Mung and Mittal, Prateek and Matthias Hein.

---

### Official Review · Reviewer_EoRx · 2022-07-11

**Rating:** 7
**Confidence:** 4
**Soundness:** 4 excellent
**Presentation:** 4 excellent
**Contribution:** 3 good

**Summary:**

The paper provides a theoretical construction in which a one hidden-layer ReLU network trained with gradient flow (under some assumptions) provably converges to a non-robust network, while at the same time a much more robust network exists. Robustness here is meant in the sense of the minimum L2 perturbations and is studied only on the training set (but this is sufficient to illustrate the point). The technical tools rely on the results from the previous works about convergence to the max-margin classifier in the parameter space which, as the authors show, can lead to the lack of robustness. This shows an interesting discrepancy between the max-margin classifier in linear models vs. one hidden-layer networks.

**Questions:**

I agree with the main point of the paper, but I wonder if the proposed theoretical constructions reflect the observed phenomena happening in practice, let’s say, for image classification. I wonder if the authors could provide some further comments on that. Some elements of the theoretical setup seem a bit artificial. For example, for a robust network in Theorem 3.1, one selects a separate weight vector for each input point which leads to only one active ReLU unit which appears to be an extreme case of memorization of the training data. Also, it seems to me that the proposed theorems (perhaps except Corollary 4.1) don’t take into account any structure of the data in the sense that they allow an arbitrary labeling of the input points on the sphere. This makes me think that the construction may be a bit artificial and not necessarily reflect some standard setting relevant to the application of neural networks in practice. It would be great to have further comments on that, in my opinion.


**Limitations:**

Perhaps the only major limitation of the analysis presented in this paper is the set of assumptions on the training data. But the main limitations are discussed quite clearly throughout the text.


---

**Update after the rebuttal.**
*I don't have any significant concerns about this paper and the questions I had were more of a speculative nature. I'm happy about the paper and the rebuttal. I think the paper is worth to be published at NeurIPS.*

**Strengths And Weaknesses:**

- **Originality.** The paper presents an interesting and novel analysis for an important problem of understanding why standard gradient-based training can lead to non-robust models, although robust models provably exist. The paper nicely points out that margin maximization in the parameter space is not aligned in general with the L2 robustness. Moreover, I think the paper is an interesting contribution towards the discussion of adversarial examples being “bugs vs. features” (https://arxiv.org/abs/1905.02175) where the authors of the current paper give theoretical evidence that adversarial examples can also arise as “bugs” from the particular optimization algorithm we use for training.
- **Quality.** High, the theory is sound, illustrated with simple-to-grasp examples, and the overall story is consistent and clear.
- **Clarity.** The paper is well-written.
- **Significance.** The previous works (Daniely and Shacham [2020], Bubeck et al. [2021], Bartlett et al. [2021]) mostly focused on analyzing the abundance of adversarial examples for random networks. However, as the authors point out, networks trained with GD are clearly not random and understanding the implicit bias of GD on robustness is thus key. Some assumptions on the input data may seem a bit restrictive (the input points are on a sphere, there is a low upper bound on the number of training points which are allowed, the network width has to be at least equal to the number of training points for the robust network) and perhaps are worth further commenting on. However, I think the paper advances our current theoretical understanding of adversarial vulnerability via the presented constructions, and it’s important that the proposed theoretical setup is well-documented and published.

---

> ### Author Response · Authors · 2022-07-31
> **Rebuttal**
>
> We thank the reviewer for the constructive feedback. To answer your questions:
>
> “Some assumptions on the input data may seem a bit restrictive…”:
>
> We will further discuss our assumptions. In a nutshell, the assumption about the input points on the sphere is made for simplicity, and we believe that it can be generalized to any points with lower- and upper-bounded norms. The upper bound on the number of training samples is due to technical reasons. We show empirically that our result holds for much more training samples and it is an interesting direction to remove this restriction
>
> “I agree with the main point of the paper, but I wonder…”
> - Regarding our construction of the robust network: Our construction indeed includes a separate weight vector for each input point. The aim of this construction is to prove that robust networks exist, and hence we constructed a simple network that shows it. There are also other robust networks that may be constructed, and it is an interesting direction to study how robust NN looks in practice.
>
> - Regarding the assumption on the structure of the data: Our assumptions allow arbitrary labels but impose restrictions on the data points. An interesting direction for future research is to relax the assumption on the data points but add assumptions on the labels. That is, perhaps assuming a certain structure on the labels may allow obtaining a similar result as ours but with weaker assumptions on the data points.
>
> - Regarding the situation in practice, e.g., in image classification: Our theoretical analysis clearly includes some simplifying assumptions. The goal of reaching a full theoretical understanding of the situation in practice is challenging and important. We believe that our result makes a step in this direction.

---

### Meta-Review · Area_Chair_Yr9p · 2022-08-28

**Recommendation:** Accept
**Confidence:** Certain

**Metareview:**

The result of the paper provides a particular vignette to the problem of lp-robustness of trained (1 hidden layer) neural networks.
The authors do not consider the statistical learning setting, but define a *robust model* essentially in terms of smoothness/robustness of the function around the *training* examples - hence not discussing generalization or function spaces (allowing arbitrary labelings). Their result states that even though a robust model (around the training samples) exists, the neural network trained with gradient descent does not find it. Or more severely, the NN solution is not robust for every training sample.

The intuition of the existence of a robust solution goes as follows (see proof sketch): Since the points are far away, we can pick each (of the m non-zero) weight roughly in the direction of each training point (s.t. only one neuron is active for one sample point) and a corresponding bias that makes sure that each training points lies "deep" inside the active region. This then leads to the network being robust (have the same prediction) around each training point in a radius of ~\sqrt{d}/2.
The lower bound for max-margin implicit bias of GD on the neural networks relies heavily on prior work by Lyu & Li '19 and Ji & Telgarsky '20.

As the reviewers agree, the story is interesting and worth publishing at Neurips, though the proof does not require significant new techniques or insights. For the camera-ready version, the authors are encouraged to add a discussion on the positioning of the insight - contrasting the current result to other works on robust *generalization*. For future work it would also be interesting to extend the results to tell a similar story in the usual learning theoretic setting.

**Award:**

No

---

### Decision · Program_Chairs · 2022-09-14

Accept